# Sensitive near point-of-care detection of asymptomatic and submicroscopic *Plasmodium falciparum* infections in African endemic countries

Dimbintsoa Rakotomalala Robinson[1,42], Ivana Pennisi[2,3,42], Matthew L. Cavuto[2,3,42], Francois Kiemde[4], Martin Chamai[5,6], Diane Yirgnur Some[4], Elliot Quigley[3], Kenny Malpartida-Cardenas[2,3], Mamadou O. Ndiath[1], Simon Correa[1], Bubacarr Darboe[1], Lindsay B. Stewart[7], Pantelis Georgiou[3,8], Mamadu Baldeh[1], Halidou Tinto[4], Aubrey J. Cunnington[9,10], Annette Erhart[1], Umberto D'Alessandro[1] & Jesus Rodriguez-Manzano[2,3,11] ✉ On behalf of the NIHR Global Health Research Group on Digital Diagnostics for African Health Systems*

Limited diagnostic capacity for detecting asymptomatic malaria infections with low parasite densities hinders elimination efforts in Africa. Here, we adapt a near point-of-care, LAMP-based diagnostic platform for malaria diagnosis using capillary blood. This Pan/*Pf* detection method meets the Malaria Eradication Research Agenda (malERA) criteria for community-level screening, with a limit of detection of 0.6 parasites/µL and a sample-to-result time under 45 minutes. We evaluate its performance on 672 capillary blood samples collected at the community level in The Gambia and Burkina Faso, including 146 *Plasmodium falciparum* positives confirmed by qPCR. The diagnostic platform achieved 95.2% sensitivity (95% CI: 90.4–98.1) and 96.8% specificity (95% CI: 94.9–98.0). It also detected 94.9% (130/137) of asymptomatic infections and 95.3% (41/43) of submicroscopic cases (<16 parasites/µL), outperforming expert microscopy (70.1% and 0%) and rapid diagnostic tests (49.6% and 4.7%). This field-deployable molecular diagnostic method offers a sensitive, scalable solution to support test-and-treat strategies for malaria elimination across Africa.

Despite substantial investment, malaria remains a significant global health concern, with an estimated 263 million cases and 597,000 deaths reported in 2023[1]. The African continent bears the highest burden, accounting for 94% of malaria cases and 95% of deaths, predominantly caused by *Plasmodium falciparum (P. falciparum)*[1]. While high transmission settings rely on universal coverage of standard interventions such as insecticide-treated nets, indoor residual spraying

(IRS), intermittent preventive treatment for pregnant women and seasonal malaria chemoprevention, regions with declining transmission urgently require innovative strategies to accelerate malaria elimination[2-4].

To further progress, it is crucial to target the entire human reservoir of infection, including both symptomatic and asymptomatic *Plasmodium*-infected individuals[5]. Passive case detection at health

---

A full list of affiliations appears at the end of the paper. *A list of authors and their affiliations appears at the end of the paper.

✉e-mail: j.rodriguez-manzano@imperial.ac.uk

facilities identifies clinical malaria cases while asymptomatic carriers in the community are unlikely to seek medical care and remain undetected. In contrast, active detection interventions (ADIs), involving community-level test-and-treat strategies, offer a promising approach to addressing this gap[5,6]. ADIs also have the advantage of avoiding the widespread drug exposure caused by Mass Drug Administration (MDA), a strategy that treats entire populations regardless of the infection status. MDA can lead to several challenges, including risk of drug resistance, unnecessary treatment of uninfected individuals and the logistical burden of implementing large-scale treatments[7,8]. However, the success of ADIs hinges on the availability of highly sensitive diagnostic tools capable of detecting all or most malaria infections[2,8,9].

Asymptomatic carriage is often associated with low parasite densities, typically below 100 parasites/μL and frequently falls under the detection threshold of conventional diagnostic methods such as light microscopy (LM) and rapid diagnostic tests (RDTs)[5,10,11]. These infections, termed submicroscopic or sub-patent, occur across the whole spectrum of malaria transmission intensity with the highest proportion (60–70%) among infected individuals in low-transmission settings[12]. Despite their low density, submicroscopic infections contribute to ongoing transmission by harbouring gametocytes that can infect mosquitoes[13,14]. RDTs and LM have a limit of detection (LOD) around 100–200 and 50–100 parasites/μL, respectively[15]. Both tests face challenges in reliably identifying asymptomatic carriers, particularly those with low parasitaemia. LM could be subjective depending on the skills of the microscopist, whereas the increased number of reported deletions of *PfHRP2* and *PfHRP3* genes is compromising the use of HRP-based RDTs[1]. In contrast, polymerase chain reaction (PCR) and quantitative PCR (qPCR) can detect infections at densities as low as 0.002 parasites/μL, but their widespread adoption faces significant challenges due to the complexity of execution, particularly in resource-constrained settings[16,17]. PCR-based techniques, including reverse-transcriptase PCR (RT-PCR)[18], require labour-intensive procedures, substantial costs, advanced laboratory infrastructure and highly skilled technicians. Additionally, results can take several hours or even days to produce, significantly delaying diagnosis and treatment[17]. Therefore, there is a pressing need for a diagnostic technology that combines both the high analytical sensitivity of molecular methods with the practicality and accessibility required for widespread use in resource-constrained field settings currently achieved by RDTs. As alternatives to standard PCR and RT-PCR, several nucleic acid amplification tests (NAATs) have emerged, including nucleic acid sequence-based amplification (NASBA), recombinase polymerase amplification (RPA), loop-mediated isothermal amplification (LAMP) and reverse transcriptase LAMP (RT-LAMP)[19–23].

LAMP-based technologies offer a promising alternative, combining the high sensitivity of molecular diagnostics with simpler equipment and operational requirements[20,24]. Unlike PCR, LAMP allows the amplification of target nucleic acid sequences at a constant temperature, an advantageous feature for field deployment, enabling the use of less expensive and more portable battery-powered block heaters. Unfortunately, similar to PCR, to achieve sufficient sensitivity LAMP requires high quality nucleic acid extraction to be performed prior to amplification[20]. As a result, many LAMP assays still rely on lengthy, multi-step nucleic acid extraction kits, making them less practical for community-based screening that demands high throughput and a shorter time-to-result[25–27]. Moreover, liquid LAMP reagents, as with reagents used in other molecular approaches, require cold-chain storage which represents a challenge for deployment in resource-constrained settings. To our knowledge, only two LAMP-based malaria diagnostic platforms are commercially available, the Loopamp™ Malaria Detection Kits (Eiken Chemical Co., Tokyo, Japan)[28] and the Alethia® Malaria platform (Meridian Bioscience Inc., Cincinnati, OH, USA), previously called Illumigene®[29]. While both platforms eliminate the need for a cold chain through lyophilisation of the

LAMP reagents, they also employ shortened sample preparation processes, which may increase reaction inhibition and lower nucleic acid recovery. Both platforms also rely on instruments that measure turbidimetry or fluorescence emission for result readout, increasing the cost and bulk of each solution[27]. In recent years, the Alethia® Malaria assay has been widely deployed in non-endemic high-income countries for the diagnosis of malaria in returning travellers, due to its high diagnostic accuracy relative to RDTs[30–32]. However, while the Alethia® platform maintains relative ease-of-use when compared to standard laboratory-based molecular methods, the cost of the instrument (>$20,000) and limited throughput (<10 samples per instrument every 40 min) remain significant barriers to its wider adoption and remote deployment in rural African settings.

To address the challenges outlined above, here we present a near point-of-care (POC) molecular approach for detecting *Plasmodium* genes. This solution, combining the magnetic bead-based nucleic acid extraction technology from SmartLid™ and the lyophilised colourimetric LAMP chemistry from the Dragonfly™ platform (originally developed for detecting viral respiratory and skin infections)[33,34], is optimised for medium to high-throughput malaria testing using capillary blood samples obtained via finger pricks in resource-limited settings. First, the analytical performance of this method was compared against the Alethia® Malaria Test, dried blood spot (DBS)-qPCR and whole blood qPCR (WB-qPCR) using *Plasmodium* culture spiked into whole blood. Next, its clinical performance was evaluated in standard laboratories at the Medical Research Council (MRC) Unit The Gambia at The London School of Hygiene & Tropical Medicine (LSHTM) and at The Clinical Research Unit of Nanoro. This was achieved by collecting capillary blood samples from individuals enroled in a community-based survey in rural Burkina Faso and The Gambia, benchmarking its accuracy against HRP2-based RDTs, LM and DBS-qPCR. A total of 672 whole blood samples were collected from 646 asymptomatic and 26 symptomatic individuals.

## Results
### Malaria detection workflow overview
An overview of the adapted Dragonfly workflow is illustrated in Fig. 1 and the entire standard operating procedure detailed in Supplementary Methods. On the front-end, an extraction method based on silica-coated superparamagnetic beads (TurboBeads™, ProtonDx) and SmartLid technology was optimised to extract parasite DNA simultaneously from up to 12 whole blood samples in under 15 min, without using a centrifuge. Lyophilised colourimetric LAMP chemistry was then used for the rapid isothermal amplification of both pan-*Plasmodium* species and *P. falciparum* targets in a single reaction well, requiring only a simple low-cost (~£100), and portable (160 × 110 × 130 mm, <1 kg), dry-bath heat block[33,35,36]. Finally, result readout and interpretation were accomplished entirely visually, with a distinct colour change from pink (negative) to yellow (positive), avoiding expensive and bulky instrumentation required for fluorescent detection. Altogether, the entire sample-to-result workflow from EDTA-anticoagulated capillary blood was accomplished for up to 12 samples within 45 min by a single user.

### SmartLid blood DNA/RNA extraction kit
SmartLid-based nucleic acid extraction technology leverages a disposable lid with a removable magnetic key to quickly and easily transfer magnetic beads and attached nucleic acids through multiple buffers and steps in the extraction and purification process. After binding nucleic acids to the silica-coated magnetic beads (Fig. 2a), collection onto the lid is performed through multiple inversions of the tube with the magnetic key inserted (Fig. 2b). The SmartLid, along with the collected magnetic beads, can then be removed from the tube and transferred into the subsequent tube. Release of the magnetic beads

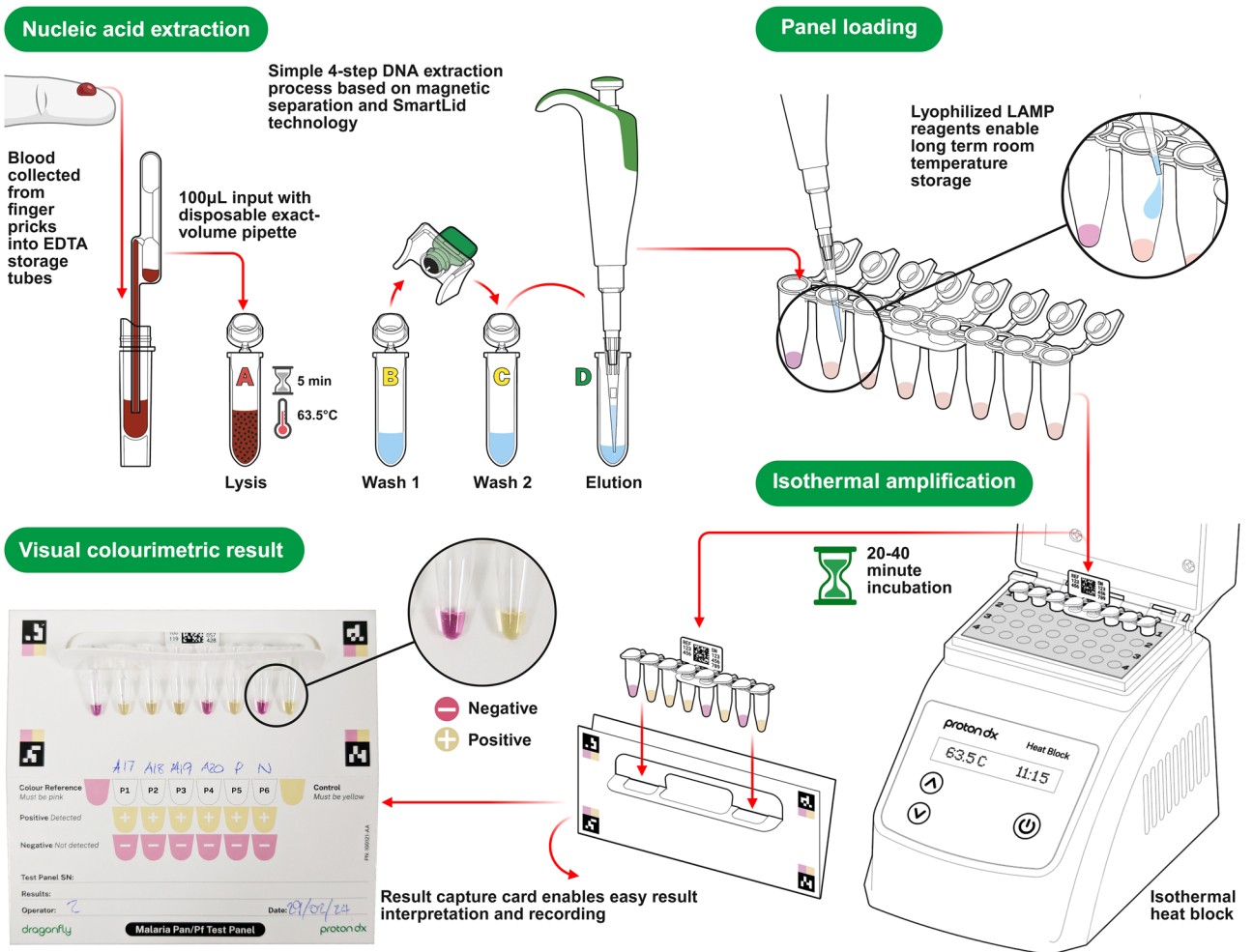

**Fig. 1 | Schematic representation of our Pan/*Pf* malaria workflow.** The integrated system combines the SmartLid whole blood extraction process with LAMP-based isothermal amplification and colourimetric readout. The SmartLid extraction process for malaria detection from finger prick whole blood involves a four-step workflow (Lysis, Wash 1, Wash 2 and Elution) including a 5-min heat-activated enzymatic incubation, and enables DNA purification and elution in under 10 min for a single sample. A total of 20 μL of eluted DNA is transferred using a fix-volume pipette into each reaction tube, followed by a maximum of 40-min incubation at 63.5 °C. Upon completion, LAMP results are qualitatively assessed by visually evaluating the colour change within the tubes, a pink colour indicating a negative result, while yellow a positive result. The validity of the test is confirmed by verifying that all controls exhibit the expected colour changes. Created in BioRender. Cavuto, M. (2025) https://BioRender.com/rwbh4tn.

into the new buffer is accomplished by removing the magnetic key and briefly shaking the tube. This entire process, transferring magnetic beads from one tube to another, is illustrated in Fig. 2c. Note that while the clear plastic (polypropylene) lid component of SmartLid is disposable, the green magnetic key is reusable, cutting down on plastic and rare-earth waste.

The originally developed SmartLid protocol to extract viral DNA and RNA from swabs stored in guanidinium thiocyanate-based buffer, eNAT® transport/inactivation media (Copan, Italy), was adapted for this study to extract human and *Plasmodium* genomic DNA from 100 μL of EDTA-anticoagulated whole blood. Notably, a 5-min heat-activated (65 °C) enzymatic (proteinase K) lysis step was added to help break down the protein-rich sample matrix, along with an additional wash step to reduce contaminant carry-over into the elution. Finally, vortex mixing was utilised instead of manual shaking to agitate the magnetic beads in each extraction buffer, which was encouraged due to the higher viscosity and more inhabitant-rich sample matrix when compared to typical respiratory or skin swab eluent. For a single sample, the entire extraction process can be completed in approximately 10 min. All kits used for the study (SmartLid Blood DNA/RNA Extraction Kit) were custom produced in collaboration with ProtonDx Ltd (https://www.protondx.com/).

## SmartLid adaptation for medium-high throughput sample processing

The original SmartLid extraction method was developed for use at the POC and assumed processing only a single sample at a time with single use cardboard trays utilised both as the kit component packaging and as a workstation. For this study, as shown in Fig. 3, the SmartLid method was adapted for medium-high throughput sample processing by developing two key tools which together enabled the simultaneous processing of up to 12 samples at a time by a single user. First, a 3D printed (X1 Carbon with AMS, Bambu Lab) tube rack was created, with 12 columns (one for each sample) of four rows (one for each step in the sample extraction process), labelled 1-12 and A-D for the columns and rows respectively (Fig. 3a). The spacing of each column and row was optimised to enable easy transfer of SmartLids from tube to tube within a column without obstruction from the open flip-cap lids. Users were also encouraged to write the column number on top of each SmartLid to further reduce the likelihood of accidentally mixing up two samples. Next, a multi-tube vortex tool was developed to conveniently hold up to 12 sample tubes (with attached SmartLids), enabling simultaneous mixing (Fig. 3b–f). The central column on the underside of the vortex tool is depressed into the centre of the vortex mixer, while a screw-on lid

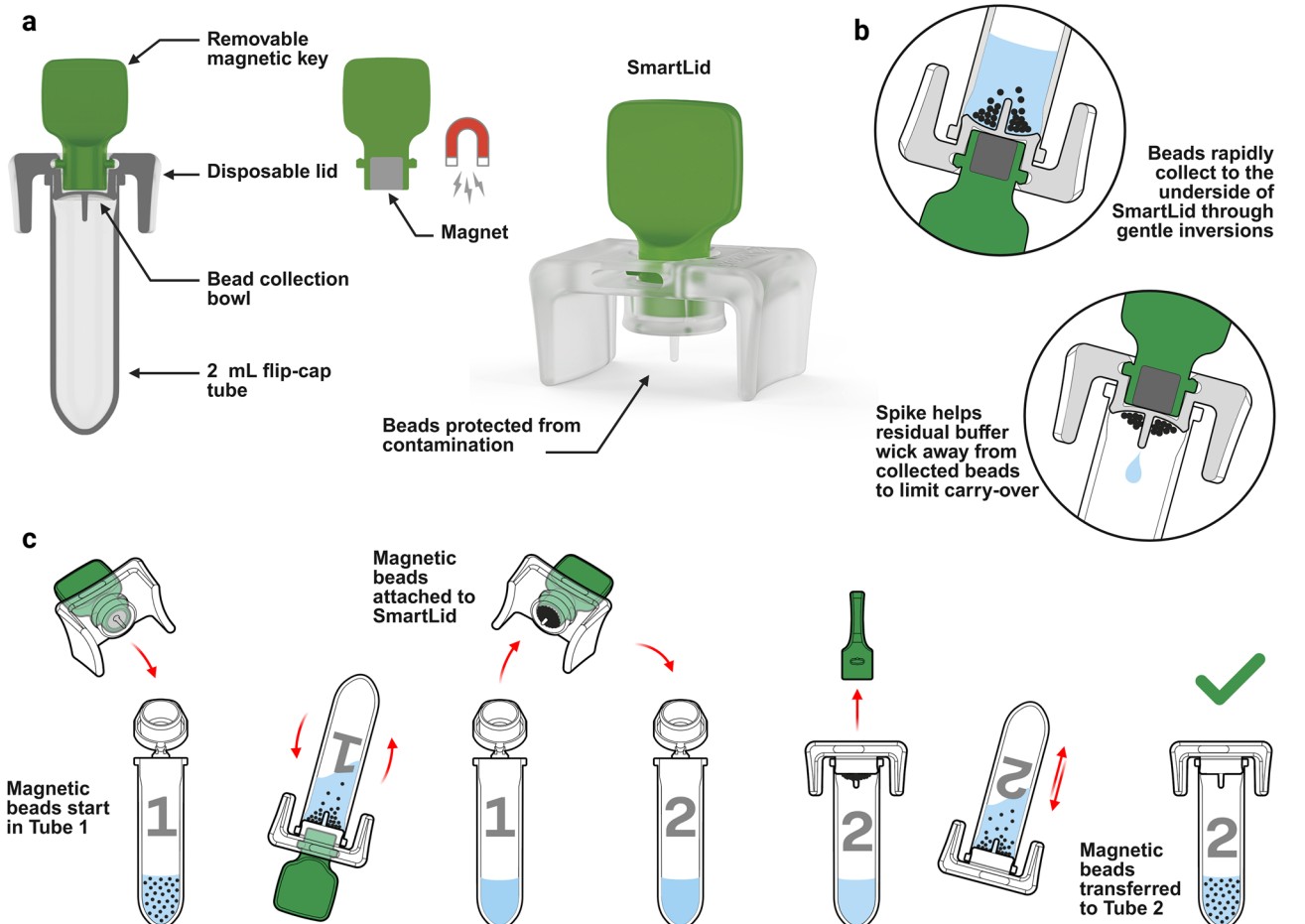

**Fig. 2 | Illustrated overview of SmartLid technology. a** SmartLid is composed of two main components, a disposable clear plastic lid, designed to press-fit into most 2 mL flip-cap or screw-cap tubes, and a removable magnetic key, housing a 5 mm × 5 mm N42 neodymium magnet. **b** Magnetic beads are collected onto SmartLid when the magnetic key is inserted, and the tube is inverted. A fluid wicking spike on the underside of the lid reduces buffer carry-over from tube to tube. **c** The entire magnetic beads collection, transfer, and resuspension process is illustrated, which occurs multiple times throughout the SmartLid extraction process. Created in BioRender. Cavuto, M. (2025) https://BioRender.com/rwbh4tn.

with a handle clamps down on the top of each SmartLid. Finally, each tube location in the tool is numbered to allow easy correlation with the number on each SmartLid and/or column number in the tube rack. Combined, these two tools enabled 12 whole blood samples to be extracted in parallel in under 15 min by a single user.

## Multi-patient malaria Pan/*Pf* test panel design

The Pan/*Pf* malaria test panel presented here was adapted from the single-patient, multi-pathogen format to a format designed to test multiple patients for a single pathogen in order to increase throughput and reduce cost per patient[34]. Each flip-cap tube within the 8-tube strip panel contained lyophilised colourimetric LAMP reagents which are stable at room temperature for extended periods and provide a clear visual colour change from pink to yellow to indicate a positive result. The tandem Pan/*Pf* assay was designed to target both pan-*Plasmodium* DNA sequences, conserved across multiple *Plasmodium* species responsible for malaria, and *P. falciparum*-specific sequences. Primers specifically targeting *P. falciparum* were included in the reaction to improve the overall sensitivity of the test, given that *P. falciparum* is the most prevalent species in the region where the study was conducted. A comprehensive list of the previously published primer sequences used in this study is provided in Supplementary Table 1[21,37].

In addition to six Pan/*Pf* target reactions, allowing for the simultaneous screening of six individuals, two control reactions were included to ensure validity of results. First, as the reaction utilises an unbuffered LAMP system and pH sensitive dye to detect amplification, a colour reference control reaction was included, which omits polymerase enzymes to prevent amplification. This reaction will always remain pink with the exact shade of pink varying based on the starting pH of the eluted nucleic acids. Depending on the sample type (e.g. capillary blood from finger pricks versus dried blood spot eluates), this starting pH and subsequent colour can vary slightly. Therefore, this reaction provides a reference colour against which all other reaction outcomes are compared. An internal control reaction was also included, targeting an exogenous DNA template lyophilised with the rest of the reaction reagents, which amplifies if the correct incubation temperature was reached, the reaction was incubated for sufficient time and reagents were not damaged in storage or expired.

Reaction positions in the 8-tube strip, from left to right, were as follows: colour reference control (tube 1), six independent Pan/*Pf* reactions (tubes 2–7), and the internal control (tube 8). Each reaction was reconstituted with 20 μL of sample eluate. Isothermal amplification was conducted using a portable dry bath heat block (ProtonDx, UK) set at 63.5 °C. Finally, while the heat block was powered by mains electricity in this study, it is also compatible with portable batteries, standard 12-volt supplies, or solar panels. It requires less than 20 W of continuous power to maintain the set temperature, supporting its suitability for decentralised testing in resource-limited settings.

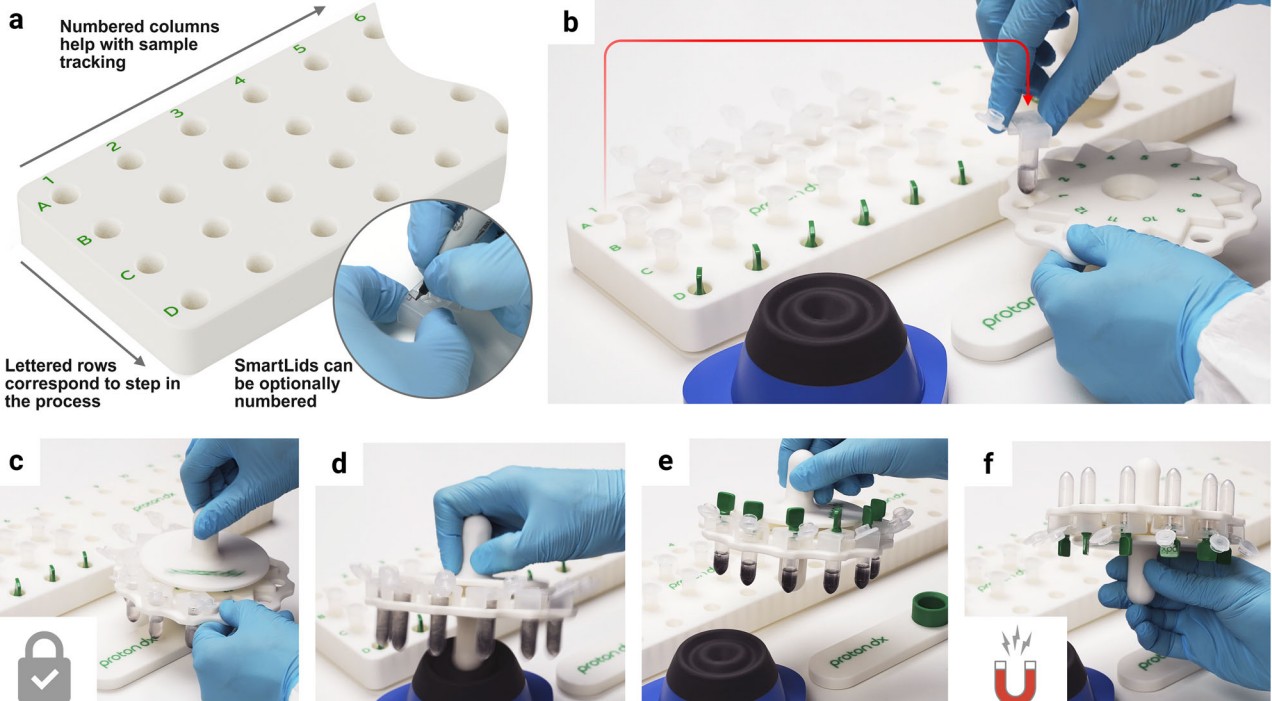

**Fig. 3 | Summary of SmartLid accessories to enable medium-high throughput sample processing. a** The SmartLid Rack, with numbered columns (1–12) to identify the sample and lettered rows (A–D) for each step in the extraction process. **b** Tube being transferred from the SmartLid Rack into the SmartLid Vortex Tool by a user performing six simultaneous extractions. **c** Screw-on plate locks all tubes and maintains all SmartLids in place while also providing a handle. **d** All samples are mixed simultaneously and equally by depressing the top of the vortex mixer with the central column underneath the tool. **e** All tubes are fully mixed and magnetic keys are inserted into each SmartLid. **f** All magnetic beads are collected simultaneously through inverting the Vortex Tool. Created in BioRender. Cavuto, M. (2025) https://BioRender.com/rwbh4tn.

### Assessment of analytical specificity and incubation time

To assess the risk of false positives, the analytical specificity of our test was evaluated at different incubation time points up to 60 min. At 50 min, the test demonstrated a specificity of 98.3% (95% CI: 96.0–100%), with 2 false positive results out of 120 negative samples tested, as shown in Supplementary Table 2. A maximum incubation time of 40 min was required to confirm negative results, although most positive reactions turned yellow within 20 min.

### Comparison of analytical sensitivity using in vitro cultured ring stages (3D7 strain)

Using serial dilutions of 3D7 parasite cultures, both LAMP platforms, Dragonfly (input volume 100 μL) and Alethia® (input volume 50 μL), consistently detected all replicates down to a parasitaemia of 0.9 parasites/μL, outperforming DBS-qPCR (input equivalent to 3 discs of 3 mm diameter, equal to ~9 μL blood), which showed decreased sensitivity below 3.8 parasites/μL. WB-qPCR (input volume 100 μL) demonstrated the highest analytical sensitivity among the four molecular-based methods (Fig. 4a). The LODs (parasites/μL) estimated by probit analysis were 2.9 [95% CI: 1.8–4.8], 0.7 [95% CI: 0.3–1.3], 0.6 [95% CI: 0.3–1.4] and 0.4 [95% CI: 0.2–0.6] for DBS-qPCR, Alethia®, Dragonfly and WB-qPCR, respectively (Fig. 4b).

### Validation of the Dragonfly platform against RDTs and LM using DBS-qPCR as the gold standard

First, a total of 50 capillary blood specimens from febrile malaria patients, purposely selected as positive controls based on concordant *Plasmodium* detection by LM, RDTs and DBS-qPCR, were also tested using the Dragonfly platform. All 50 samples tested positive, confirming the compatibility of our method with real, field-collected capillary blood samples prior to evaluation on unlabelled community survey samples. Next, 672 capillary blood samples collected at the community-level in The Gambia and Burkina Faso were used to evaluate the performance of the Dragonfly platform against DBS-qPCR as the reference method. All samples were also assessed using expert LM and RDTs to benchmark the performance of our method against the two standard diagnostic approaches for malaria. Of the 672 samples, 27.1% (146/672) were positive for *P. falciparum* by DBS-qPCR. These positive samples represented a broad range of parasite densities, including both microscopically detectable (*n* = 103) and submicroscopic infections (*n* = 43). A breakdown of the sample categories is presented in Fig. 5. Detailed results obtained using Dragonfly, LM, RDTs and DBS-qPCR for each of the 50 positive controls from the confirmed malaria patients, as well as for the 672 capillary blood samples collected at the community level, are provided as Supplementary Data 6.

As depicted in Fig. 6, considering the 672 samples, the overall sensitivity and specificity of our method against DBS-qPCR was 95.2% [95% CI: 90.4–98.1] and 96.8% [95% CI: 94.9–98.0], respectively. Comparatively, the Dragonfly method demonstrated a higher sensitivity than both RDT (50.7% [95% CI: 42.3–59.0]) and expert LM (70.5% [95% CI: 62.4–77.8]). All three methods achieved specificities above 96%, with expert LM recording the lowest false-positive rate.

When considering samples from asymptomatic individuals only (*N* = 646, including 137 malaria positive and 509 negative samples as determined by DBS-qPCR) the sensitivity gap remained similar, with Dragonfly, LM and RDTs detecting 94.9%, 70.1% and 49.6% of positive samples, respectively. Confusion matrices for this subset of samples, as well as the smaller subset of symptomatic cases, are provided as Supplementary Fig. 1.

**a**

| Parasitemia (p/µL) | DBS-qPCR POS/TOTAL (%) | Alethia® POS/TOTAL (%) |
|---|---|---|
| 6000 | 10/10 (100) | 5/5 (100) |
| 600 | 10/10 (100) | 5/5 (100) |
| 60 | 10/10 (100) | 5/5 (100) |
| 30 | 10/10 (100) | 5/5 (100) |
| 15 | 10/10 (100) | 5/5 (100) |
| 7.5 | 10/10 (100) | 5/5 (100) |
| 3.8 | **10/10 (100)** | 5/5 (100) |
| 1.9 | 7/10 (70) | 5/5 (100) |
| 0.9 | 4/10 (40) | **10/10 (100)** |
| 0.5 | 0/10 (0) | 9/10 (90) |
| 0.25 | 0/10 (0) | 6/10 (60) |
| 0.125 | 0/10 (0) | 4/10 (40) |

| Parasitemia (p/µL) | Dragonfly POS/TOTAL (%) | WB-qPCR POS/TOTAL (%) |
|---|---|---|
| 6000 | 5/5 (100) | 10/10 (100) |
| 600 | 5/5 (100) | 10/10 (100) |
| 60 | 5/5 (100) | 10/10 (100) |
| 30 | 5/5 (100) | 10/10 (100) |
| 15 | 5/5 (100) | 10/10 (100) |
| 7.5 | 5/5 (100) | 10/10 (100) |
| 3.8 | 5/5 (100) | 10/10 (100) |
| 1.9 | 5/5 (100) | 10/10 (100) |
| 0.9 | **10/10 (100)** | 10/10 (100) |
| 0.5 | 9/10 (90) | **10/10 (100)** |
| 0.25 | 7/10 (70) | 8/10 (80) |
| 0.125 | 5/10 (50) | 3/8 (37.5) |

**b**

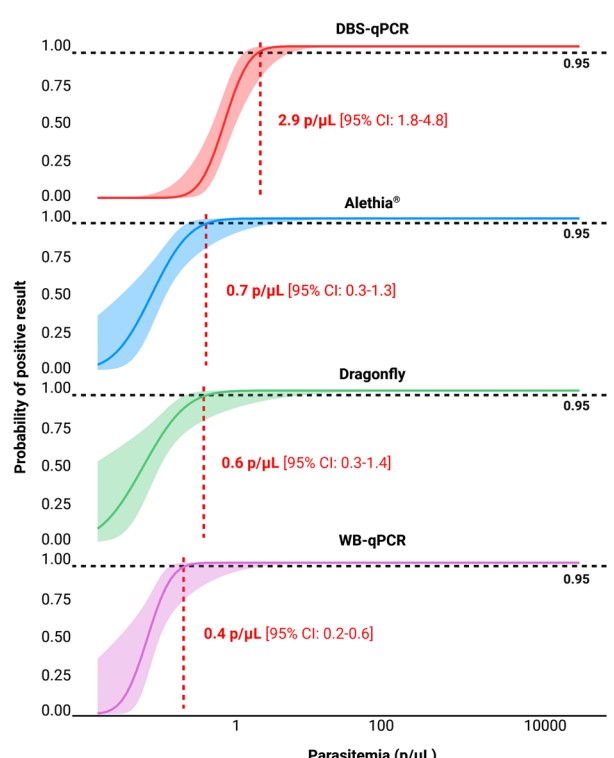

**Fig. 4 | Comparison of analytical sensitivity of malaria detection using spiked whole blood across Dragonfly, Alethia®, DBS-qPCR, and WB-qPCR.** Experiments were conducted in collaboration with LSHTM using spiked EDTA-blood with ring-stage *P. falciparum* 3D7 strain. Results are shown in terms of **a** number and percentage of successfully detected replicates at each spike concentration as well as **b** the resulting empirically determined LOD through probit analysis. The

solid curves represent the predicted probabilities of positive results as a function of parasite density, with shaded areas indicating 95% CIs. The red dashed line denotes the LOD or the parasite density at which the probability of a positive result is 95%. Created in BioRender. Cavuto, M. (2025) https://BioRender.com/rwbh4tn. Source data are provided as a Source Data file.

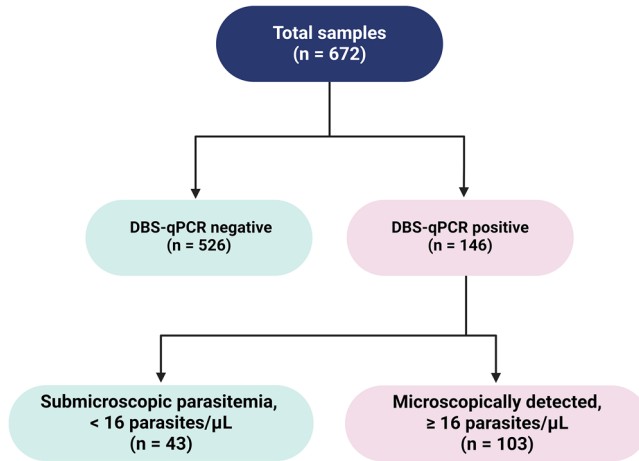

**Fig. 5 | Clinical sample selection to evaluate the performance of Dragonfly Pan/*Pf* malaria platform.** Submicroscopic parasitaemia is defined as a parasite density of <16 parasites/µL, corresponding to the theoretical LOD for an expert microscopist, based on the ability to detect one asexual parasite among 500 leucocytes, assuming a white blood cell count of 8000 leucocytes/µL. Created in BioRender. Cavuto, M. (2025) https://BioRender.com/rwbh4tn. Source data are provided as a Source Data file.

As summarised in Fig. 7, the 146 DBS-qPCR positive specimens were stratified into four parasite density groups: <16 parasites/µL ($n$ = 43), 16–100 parasites/µL ($n$ = 28), 100–200 parasites/µL ($n$ = 13) and >200 parasites/µL ($n$ = 62). Across the three lower density

categories, Dragonfly outperformed RDT, detecting 41 of 43 specimens in the <16 parasites/µL group (95.3%), 25 of 28 specimens in the 16–100 parasites/µL group (89.3%), and all 13 specimens in the 100–200 parasites/µL group (100%). In contrast, RDT detected 2 of 43 specimens (4.7%), 9 of 28 specimens (32.1%), and 8 of 13 specimens (61.5%) in the corresponding groups respectively. Dragonfly also demonstrated significantly higher sensitivity than expert LM for submicroscopic parasitaemia (<16 parasites/µL); however, no significant differences ($p > 0.05$) were observed between Dragonfly and expert LM in the 16–100 and 100–200 parasites/µL groups. When parasite density exceeded 200 parasites/µL, all three methods showed comparable performance ($p > 0.05$).

## Discussion

Through collaborative efforts between UK and West African institutions, this study presented a near-POC colourimetric LAMP-based extracted molecular solution to achieve accurate detection of submicroscopic *Plasmodium* infections from whole blood at the community level in Sub-Saharan Africa. Our approach demonstrated high analytical performance, achieving an LOD of 0.6 parasites/µL [95% CI: 0.3–1.4] with spiked samples. Field evaluation demonstrated a sensitivity and specificity of 95.2% [95% CI: 90.4–98.1%] and 96.8% [95% CI: 94.9–98.0%] respectively from individuals enrolled at community level, most of them (96%) asymptomatic. This high diagnostic accuracy, along with the ability to detect 95.3% of the submicroscopic infections (<16 parasites/µL), most of which were missed by both RDTs and LM, suggests that the approach may be considered a valuable tool for community-based ADIs in malaria-endemic regions. Moreover, all readings in our study were performed by expert microscopists to

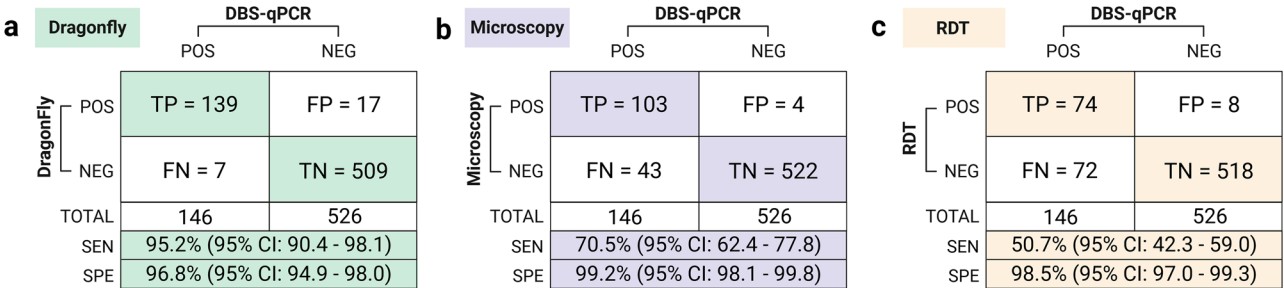

**Fig. 6 | Clinical performance comparison of malaria diagnostic methods used in this study.** Comparison of the clinical performance of Dragonfly (**a**), LM (**b**) and RDTs (**c**) using whole blood finger prick samples, with DBS-qPCR as the gold-standard comparator. For each group, the number of true positives (TP), total positive cases, sensitivity rate with 95% CI, number of true negatives (TN), total negative cases, and specificity rate with 95% CI are provided. FP indicates the number of false positives, FN indicates the numbers of false negatives. Created in BioRender. Cavuto, M. (2025) https://BioRender.com/rwbh4tn. Source data are provided as a Source Data file.

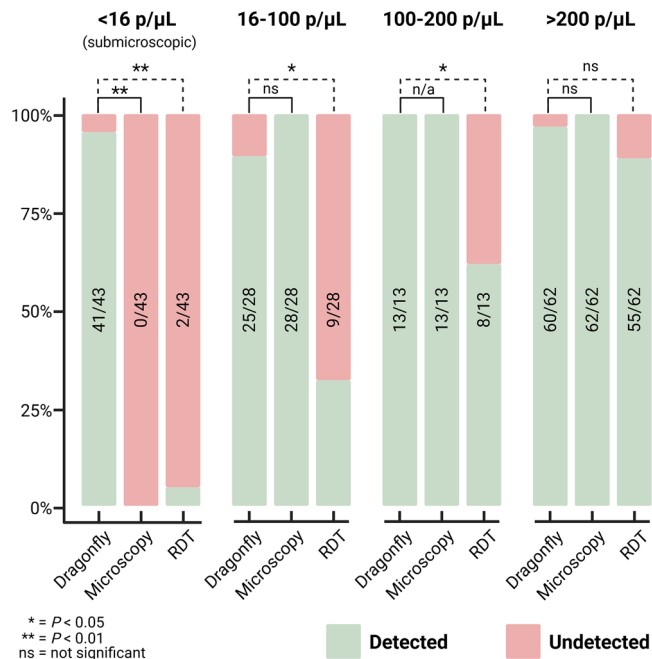

\* = P < 0.05
\*\* = P < 0.01
ns = not significant

**Fig. 7 | Detection of DBS-qPCR positive samples by Dragonfly, LM and RDTs, stratified by four parasite density categories: <16, 16–100, 100–200 and >200 parasites/μL.** For each category, two-sided McNemar's tests were applied to compare paired diagnostic outcomes between Dragonfly and LM, and Dragonfly and RDTs. Dragonfly correctly identified 95.3% of submicroscopic infections, 41 out of 43 samples detected with <16 parasites/μL, significantly outperforming both expert LM ($p < 0.0001$) and RDTs ($p < 0.0001$). Dragonfly also significantly outperformed RDTs ($p = 0.0001$) at densities ranging from 16–100 parasites/μL, while no significant difference was observed between Dragonfly and expert LM ($P = 0.0833$). Similarly, for the 100–200 parasites/μL range, Dragonfly significantly outperformed RDTs ($p = 0.0253$), while detecting the same number of samples as expert LM. Finally, at parasite densities >200 parasites/μL, no statistically significant differences were observed between Dragonfly and either expert LM ($p = 0.1573$) or RDTs ($p = 0.0956$). Created in BioRender. Cavuto, M. (2025) https://BioRender.com/rwbh4tn. Source data are provided as a Source Data file.

ensure accurate identification of parasites, especially for low parasite density infections. This rigorous approach likely contributed to the higher sensitivity by LM observed in our study compared to routine clinical practice, where blood slides may be read by less experienced technicians. Furthermore, despite the comparable performance of LM for parasite densities above 16 parasites/μL demonstrated through this study, multiple practical limitations remain that make LM unsuitable for large-scale asymptomatic screening. For example, the preparation and reading of a slide, particularly at low parasite densities, can take an experienced microscopist at least 25–30 min per sample. In contrast, the presented method can deliver results for up to 12 samples simultaneously in an average of 45 min per minimally trained operator.

Since the introduction of LAMP technology for *Plasmodium* detection, more than 26 LAMP assays have been developed and evaluated, demonstrating an estimated pooled sensitivity of 97.1% [95% CI: 95.7–98.0%] as reported in a previous meta-analysis study including both symptomatic and asymptomatic individuals[38]. However, to date only two LAMP-based diagnostic tests (Loopamp™ Malaria and Alethia® Malaria) are currently commercialised for malaria, suggesting that high technical performance alone is insufficient to ensure the sustainable deployment of a diagnostic test in its intended target settings[20,26]. Performance studies conducted in both health facilities and community settings demonstrated varying sensitivities of these two commercial options, ranging from 98.4% to 40.8% for symptomatic and asymptomatic case, respectively[39–43]. In our study, Alethia® demonstrated comparable high analytical performance, which combined with its user-friendliness likely contributes to its widespread adoption in high-income countries to guide malaria diagnosis in returning travellers[30,32]. However, its deployment in rural African settings, especially for large-scale community-based malaria screening interventions is constrained by its high cost and the limited capacity of the Alethia incubator accommodating only 10 samples per run. In contrast, Dragonfly combines high diagnostic accuracy with several key advantages for field deployment in resource constrained settings, meeting all the essential (and many of the desirable) technical and health systems criteria outlined in the malERA Target Product Profiles for diagnostics intended for malaria screening and surveillance, as detailed in Supplementary Table 3[17]. These specifications include ease of sample collection, high sensitivity and specificity, rapid turnaround time, ease of use and portability. For example, the SmartLid DNA extraction method is compatible with capillary finger prick whole blood, enabling field sampling using well established procedures (finger prick blood samples) in routine healthcare and malaria testing across Sub-Saharan Africa. Furthermore, the SmartLid technology and optimised protocol ensures quality nucleic acid extraction and purification in a fraction of the time (<15 min for up to 12 samples simultaneously) of gold-standard extraction methods, which can take well over an hour and often rely on bulky and expensive centrifuges. On the detection side, the lyophilised isothermal colourimetric LAMP chemistry forgoes the cost and bulk of thermocyclers and devices that rely on LED illuminators or fluorescent detectors to determine the result. Altogether, 12 whole blood samples can be extracted and amplified from start to finish in as little as 35 min for high parasitaemia samples, relying only on a vortex mixer (optional) and two low-cost isothermal heat blocks for powered equipment, and room temperature storage for all consumables. Further comparison of the characteristics of the Dragonfly platform with the two commercial malaria LAMP technologies is shown in Supplementary Table 4.

ADIs such as Mass-Testing-and-Treatment (MTAT) and Focused Testing and Treatment are currently not recommended by WHO due to their limited or negligible impact on malaria prevalence and incidence of clinical malaria[44,45]. However, such recommendations are primarily based on intervention trials that used RDTs and/or LM for malaria diagnosis[44,45]. Recent modelling studies suggest that deploying a diagnostic test with sufficiently high sensitivity, such as one that reduces the LOD below 200 parasites/μL, can accelerate malaria elimination, provided a high coverage of sufficient duration is achieved, and the treatment is efficacious[8,9,46]. It has been further shown that by reducing the LOD below 2 parasites/μL, MTAT strategies could substantially increase the identification of submicroscopic cases, leading to a reduction in *P. falciparum* PCR-based prevalence and a decrease in the required number of intervention rounds[9].

Since our platform is still in a prototype development stage, a comprehensive cost analysis of the final manufactured version is currently unavailable, but will be a significant factor in the assay's suitability for deployment in sub-Saharan Africa. However, a preliminary cost analysis is provided in Supplementary Table 5, based on prototype quantities and estimations at low- and medium-scales, and is compared to the current market costs of the Alethia® Malaria Test and associated required instrument. Another limitation of our study is that Dragonfly malaria testing was performed in standard laboratories in The Gambia and Burkina Faso, which do not reflect the real-life conditions of field sampling and testing. Therefore, future studies will assess the robustness of the device in more decentralised environments such as community settings and gather detailed insights into user experiences and device usability in the field. For example, studying the practical shelf-life of the test kits at a realistic range of room-temperatures throughout the continent will be critical. Current prototype Dragonfly test panels are conservatively labelled to expire after at least 6 months at 30 °C, while the SmartLid Blood DNA/RNA Extraction Kits remains stable for at least 12 months. In preparation for this investigation, the SmartLid extraction method described here is also being adapted into the true-POC single-use format of the previously presented Dragonfly sample-to-result platform[34], with all reagents and buffers pre-aliquoted and laboratory micro-pipettes replaced with disposable exact volume pipettes. Finally, although the technology development and adaptation were performed in laboratories in London through an active collaboration with African Institutions, future initiatives should focus on extending the concept of transferability to local development and production in Africa. This approach would facilitate the sustainable manufacturing and distribution of the diagnostic platform, thereby ensuring its availability in malaria-endemic regions.

In addition to demonstrating the strong potential of the SmartLid extraction and Dragonfly colourimetric LAMP technologies towards malaria elimination strategies, this work highlighted the rapid adaptability of the combined approach and potential for assay transfer, which should be a key criterion for selecting molecular diagnostic tests in Sub-Saharan Africa[47]. Our approach's significant use of off-the-shelf consumables and flexibility enabled the rapid prototyping and deployment of the presented multi-patient Pan/*Pf* malaria test from capillary finger prick blood samples. Finally, the potential for future digital and cloud integration of the Dragonfly platform, as explored in previous works demonstrating solutions for viral respiratory and skin infection diagnostics, aligns with the growing need for connected diagnostic solutions in Africa[34,48].

## Methods
### Ethical statement
Collection of samples during community-based surveys in The Upper River Region in The Gambia was approved by the LSHTM Ethics Committee and The Gambia Government/MRC Joint Ethics Committee (ref. 29611). In Burkina Faso, sample collection from symptomatic patients enroled at health facilities in the Central West Region received approval from The Comité d'Éthique pour la Recherche en Santé-Burkina Faso (ref: DELIBERATION N°2021-04-084) and the Comité d'Éthique Institutionnel pour la Recherche en Sciences de la Santé of IRSS (N/Réf. A07-2021/CEIRES). The approval for collecting samples from participants enroled at the community level was granted by The Comité d'Éthique pour la Recherche en Santé-Burkina Faso (ref: DELIBERATION N° 2024-0361). Written informed consent was obtained from all research participants and/or guardians before recruitment.

### Comparison of analytical sensitivity using in vitro cultured ring stages (3D7 strain)
The analytical sensitivity of Dragonfly was evaluated in comparison with Alethia®, DBS-qPCR and WB-qPCR. The comparison was performed using serial dilutions of ring-stage *P. falciparum* (3D7 strain) parasite culture at the malaria parasitology lab of the LSHTM in London. Parasites were cultured in-vitro and synchronised using a magnetic separation procedure[49]. To minimise the occurrence of red blood cells (RBCs) infected by multiple parasites, cultures were maintained under gentle agitation (60RPM), resulting in 85% of singly infected RBC among the total infected cells (Supplementary Fig. 2). The final parasitaemia of the culture, measured at 5.8% of RBCs, was confirmed by expert LM. A *Plasmodium*-negative blood sample (tested by WB-qPCR) was then spiked to generate the initial infected sample, yielding a parasitaemia of 6000 parasites/μL. Serial dilutions were subsequently performed using the same negative whole blood sample to generate samples with decreasing concentrations, calculated based on the dilution factor applied. Each dilution point was tested multiple times across the four molecular-based methods. The concentrations of each dilution point, as well as the number of replicates per method, are summarised in Fig. 4.

Alethia® is a commercially available LAMP-based technology that utilises a *Plasmodium* genus-specific assay. The system consists of an initial DNA extraction process using a passive gel filtration column, followed by an amplification step with lyophilised reagents in a dedicated LAMP incubator. The test provides on an LCD screen a qualitative result (positive or negative), which is automatically interpreted by a reader integrated within the incubator. As per the manufacturer's guidelines, 50 μL of whole blood was used as the input volume in our evaluation.

DNA was extracted from DBS (3 discs of 3 mm diameter, equal to approximately 9 μL blood) and whole blood samples (100 μL) using the QIAamp DNA Mini Kit (cat. nos. 51304 and 51306) according to the manufacturer's instructions. DNA was eluted in AE Buffer with a final volume of 100 μL for DBS and 200 μL for whole blood samples. The *P. falciparum*-specific PCR assay applied to both sample types has been previously described[50]. The PCR reaction volume was 20 μL, comprising 5 μL of DNA extract, 10 μL of GoTaq qPCR Master Mix 2X (cat. no. A6101), 2 μL of 10X *P. falciparum* primers/probe mix, and 3 μL of PCR grade water. Amplification was performed on a LightCycler® using the following cycling conditions: an initial denaturation at 95 °C for 2 min, followed by 45 cycles of denaturation at 95 °C for 15 s and annealing/extension at 60 °C for 1 min.

### Dragonfly malaria field validation against RDTs and LM versus DBS-qPCR
The compatibility of the adapted Dragonfly method with finger prick clinical samples was assessed using 50 blood specimens purposively selected based on their positivity for *Plasmodium* by RDTs, LM and DBS-qPCR. These specimens were obtained from febrile patients attending rural health facilities in the Central Region in Burkina Faso. Parasite densities, determined by expert LM, ranged from 149 to 87,500 parasites/μL with a median [IQR] of 714 parasites/μL [95% CI: 128–5688 parasites/μL].

**Table 1 | Characteristics of study participants**

| Characteristics | n (%) |
|---|---|
| **Sex** | |
| Male | 289 (43.0) |
| Female | 383 (57.0) |
| **Age groups in years** | |
| 0.5–4 | 78 (11.6) |
| 5–14 | 198 (29.5) |
| 15–29 | 126 (18.7) |
| 30–59 | 188 (28.0) |
| ≥60 | 82 (12.2) |
| **Symptomatic status** | |
| Symptomatic | 26 (3.9) |
| Asymptomatic | 646 (96.1) |
| **Sample collection sites** | |
| The Gambia | 281 (41.8) |
| Burkina Faso | 391 (58.2) |

A total of 672 blood specimens were tested to evaluate the performance of the presented method compared to RDTs and LM using DBS-qPCR as gold standard. Blood specimens were collected from individuals (n = 289 males, n = 383 females, with ages spanning 0.5 to 88 years) enroled in a community-based survey in two malaria endemic sites (The Central Region in Burkina Faso and The Upper River Region in The Gambia) characterised by a predominance of *P. falciparum*. Baseline characteristics of study participants are summarised in Table 1 and show a slight predominance of females (57%) and a fair representation of all age categories. Most participants (96%) were asymptomatic at the time of sample collection. Informed consent was obtained from all participants and/or guardians prior to their enrolment.

All malaria tests were performed using capillary blood collected by finger prick. RDT testing was performed on-site, either in a health facility (50 blood specimens used for the compatibility assessment) or at community level (672 blood specimens used for the diagnostic performance evaluation). LM readings were performed on thick smears. Clinical blood specimens used for the Dragonfly method collected into 200 μL EDTA microtainers were stored between 3 and 5 °C if they were tested the same day as sample collection, or otherwise stored at –20 °C until testing. Blood spots from the same finger prick were collected on DBS cards and used for qPCR analysis.

### Malaria confirmation by DBS-qPCR

DBS-qPCR was selected as the gold standard method for evaluating the performance of the developed platform. The DBS-qPCR testing was conducted at the MRCG following a standardised protocol[51]. The gold standard method consisted of capillary blood spotted onto Whatmann filter paper (cat. no. 1600-003) followed by a qPCR detection using a TaqMan assay targeting the *var* gene acidic terminal sequence of *P. falciparum*. The number of copies of the *var* gene is approximately 59 per genome. Genomic DNA extraction was performed using the QIAamp 96 DNA QIAcube HT Kit (cat. no. /ID. 51331), in accordance with the manufacturer's instructions. Three discs of 3 mm diameter were punched out of the DBS card and the DNA extracts were eluted in a volume of 80 μL of AE Buffer. DNA amplification was conducted using the Bio-Rad CFX96 real-time PCR machine. Each reaction utilised 5 μL of sample DNA. The varATS qPCR master mix consisted of 1 μL of PCR-grade water, 10 μL of 2X TaqMan Master Mix, 1.6 μL of 10 μM *Var* forward primer, 1.6 μL of 10 μM *Var* reverse primer, and 0.8 μL of 10 μM Var probe. The cycling conditions were: 1 cycle at 50 °C for 2 min and 1 cycle at 95 °C for 10 min followed by 45 cycles at 95 °C and 55 °C for respectively, 15 s and 1 min. If the result was positive for either PCR or

Dragonfly alone, the PCR was repeated in duplicates starting from a new DNA extraction. If at least one of the repeats was positive, the sample was considered positive for *P. falciparum*.

### Light microscopy

The thick blood smear was prepared from fresh capillary blood and air dried. Giemsa staining was performed with a 3% solution for 30 min. All slides were independently read by two qualified expert microscopists. Parasite density was determined by counting the number of asexual parasites per 500 leucocytes on the thick smear, assuming a leucocyte count of 8000/μl, and using 100x magnification with optical LM. In case of discrepancy between the two readers, such as one reporting negative and the other positive for malaria, or a density difference of >50%, or different species reported, a third expert microscopist would read the slide as well. The reported parasite density was the geometric mean of the two readers' results or that of the two closest readings if a third reading was done.

### Rapid-diagnostic tests

*P. falciparum* infection status of the 672 community-collected samples was assessed using the SD BIOLINE Malaria *P. falciparum* Ag Test™ (Abott cat. nos.: 05FK50, 05FK51, 05FK52, 05FK53) or the AdvDx™ Malaria Pf Rapid Malaria Ag Detection Test (cat. no. 00-DKM-RK-MAL-ADX-004-050). Both are *HRP2*-based immunolateral flow assays accepted on the WHO list of prequalified in vitro diagnostics[52,53].

### The Dragonfly Pan/*Pf* malaria test workflow

Extractions were performed manually using the SmartLid sample preparation in bulk reagent format allowing for efficient processing of multiple samples simultaneously. A total of 100 μL of capillary blood sample was used as input for each extraction. Purified samples were eluted in a total elution volume of 50 μL. LAMP reactions were performed by adding 20 μL of the eluted sample into each tube of the colourimetric LyoLAMP Pan/*Pf* Test Panels (ProtonDx). The pan-*Plasmodium* (Pan18s) *and P. falciparum* (PfK13, Pf mtDNA) assay genes and primer sequences used in this study are listed in Supplementary Table 1. Amplification was conducted in the portable isothermal heat block (ProtonDx) for up to 40 min at 63.5 °C. Immediately following amplification, results were visually assessed by the technician based on the colour change in each tube, with a change from pink to yellow indicating a positive result. Note, intermediate colour shifts (from pink to yellow, e.g. orange) were considered as positive results, as this still indicated amplification of product.

All testing was carried out in standard molecular biology laboratories at MRC Unit The Gambia and Clinical Research Unit Nanoro, Burkina Faso. All steps, from DNA extraction to result-read-out, were conducted in a single workspace without the need for a laboratory hood. Prior to testing, a half-day training programme was conducted for users including both theoretical instructions and hands-on practice using 3D7-infected whole blood and negative controls.

### Statistics & reproducibility

The minimum required sample size was calculated using equations described by Hajian-Tilaki (2014)[54]. Based on a 5% margin of error, a 95% confidence level, and an expected sensitivity of 96% and specificity of 98%[38], at least 59 samples were required. All samples were fully anonymised prior to analysis. Data from the specificity experiments are provided in Supplementary Data 1, and analytical sensitivity results in Supplementary Data 2–5. Socio-demographics, DBS-qPCR, RDT, LM and Dragonfly data of the positive controls (n = 50) and the total clinical samples (n = 672) were merged into a unique database and statistical analyses conducted using Stata 18 (StataCorp, College Station, TX, USA) and R (R Core Team, Vienna, Austria) (Supplementary Data 6). Information on sex was recorded for each participant through self-report, with numbers and proportions presented in Table 1. Diagnostic

performance data were not disaggregated by sex as this factor was not considered relevant in the evaluation of a diagnostic method for *Plasmodium* infections intended to be broadly applicable for patients, irrespective of sex. Duplicate records were removed before conducting analyses. Summary statistics were performed using the median value and IQR for continuous variables, and proportions and 95% CIs for categorical variables. Sensitivity and specificity with their 95% CIs were calculated for Dragonfly, LM and RDT, using DBS-qPCR as the gold standard according to formulas shown in Supplementary Table 6. The LOD was estimated using probit analysis defining LOD as the concentration with a 95% probability of obtaining a positive result[55]. To assess the differences between groups, McNemar's Test was employed. A $p < 0.05$ was considered statistically significant.

All malaria assays were conducted following standardised operating procedures and using calibrated instruments, with operators blinded to the outcomes of comparative tests. Samples were tested in multiple replicates during the LOD experiments to support consistency across results.

### Reporting summary
Further information on research design is available in the Nature Portfolio Reporting Summary linked to this article.

## Data availability
All data supporting the findings of this study are available within the article and its supplementary files. Any additional requests for information can be directed to, and will be fulfilled by, the corresponding authors. Source data are provided with this paper.

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

## Acknowledgements

This research was funded by the NIHR (NIHR134694) using UK international development funding from the UK Government to support global health research (to A.J.C. and H.T.). The views expressed in this publication are those of the author(s) and not necessarily those of the NIHR or the UK government. In addition, this work was supported by the Department of Health and Social Care-funded Centre for Antimicrobial Optimisation (CAMO) at Imperial College London (to J.R.M.); the Fleming Initiative, founded in partnership between Imperial College London and Imperial College Healthcare National Health Service (NHS) Trust (to J.R.M.); the Jameel Fund for Infectious Disease Research and Innovation (to J.R.M.); the Wellcome Trust CAMO-Net programme [226691/Z/22/Z] (to J.R.M.); the Wellcome Trust Innovator Award [215688/Z/19/Z] (to P.G. and J.R.M.); and the Imperial College Research Fellowship [WDPI.G09074] (to K.M.C). The views expressed in this publication are those of the authors and not necessarily those of the NIHR or the UK Department of Health and Social Care. P.G. and J.R.M. are affiliated with the NIHR Health Protection Research Unit (HPRU) in Healthcare Associated Infections and Antimicrobial Resistance at Imperial College London in partnership with the UK Health Security Agency, in collaboration with Imperial Healthcare Partners, the University of Cambridge and the University of Warwick. A.J.C., P.G. and J.R.M. are supported in part by the NIHR Biomedical Research Centre of Imperial College NHS Trust. D.R.R and M.C. were supported by Imperial College London's Research England ISPF Institutional Support ODA grant 2024-25, administered by the Global Development Hub at Imperial College London. J.R.M and P.G. acknowledge support and funding from the Centre of Defence Pathology and jHub Med, UK Ministry of Defence.

## Author contributions

D.R.R., I.P. and M.L.C. contributed equally to the work. Study concept and design: D.R.R., I.P., M.L.C., A.E., U.D.A., H.T., A.C and J.R.M. Acquisition, analysis, or interpretation of data: D.R.R., I.P., M.L.C., F.K., M.C., D.Y.S., E.Q., K.M.C., M.O.N., S.C., B.D., L.B.S., P.G., M.B., H.T., A.J.C., A.E., U.D.A. and J.R.M. Drafting the manuscript: D.R.R., I.P., M.L.C., J.R.M. Critical revision of the manuscript: D.R.R., I.P., M.L.C., F.K., M.C., D.Y.S., E.Q., K.M.C., M.O.N., S.C., B.D., L.B.S., P.G., M.B., H.T., A.J.C., A.E., U.D.A. and J.R.M. The manuscript was written through the contributions of all

authors. All authors have given approval to the final version of the manuscript.

## Competing interests

The authors declare the following competing financial interest(s): I.P., M.L.C., E.Q., K.M.C., P.G. and J.R.M. have financial interest on ProtonDx Ltd, which currently has exclusive license to intellectual property linked to Dragonfly (WO2023131803A1) and SmartLid (WO2022180376A1), and their associated trademarks. These authors declare that they do not have any other known competing financial interests or personal relationships that could have appeared to influence the work reported in this paper. The remaining authors declare no competing interests.

## Additional information

[1]MRC Unit The Gambia at the London School of Hygiene and Tropical Medicine, Fajara, The Gambia. [2]Department of Infectious Disease, Imperial College London, London, UK. [3]ProtonDx, Translation & Innovation Hub, Imperial College London, London, UK. [4]Institut de Recherche en Sciences de la Santé, Clinical Research Unit of Nanoro, Nanoro, Burkina Faso. [5]West African Centre for Cell Biology of Infectious Pathogens, University of Ghana, Accra, Ghana. [6]Department of Biochemistry, Cell and Molecular Biology, University of Ghana, Accra, Ghana. [7]Faculty of Infectious and Tropical Diseases, London School of Hygiene & Tropical Medicine, London, UK. [8]Department of Electrical and Electronic Engineering, Imperial College London, London, UK. [9]Section of Paediatric Infectious Disease, Department of Infectious Disease, Imperial College London, London, UK. [10]Centre for Paediatrics and Child Health, Imperial College London, London, UK. [11]The Fleming Initiative, Imperial College London and Imperial College Healthcare NHS Trust, London, UK. [42]These authors contributed equally: Dimbintsoa Rakotomalala Robinson, Ivana Pennisi, Matthew L. Cavuto. ✉e-mail: j.rodriguez-manzano@imperial.ac.uk

## the NIHR Global Health Research Group on Digital Diagnostics for African Health Systems

Dimbintsoa Rakotomalala Robinson[1,42], Ivana Pennisi[2,3,42], François Kiemde[4], Martin Chamai[5,6], Kenny Malpartida-Cardenas [2,3], Lindsay B. Stewart[7], Pantelis Georgiou[3,8], Mamadu Baldeh [1], Halidou Tinto[4], Aubrey J. Cunnington[9,10], Annette Erhart[1], Umberto D'Alessandro[1], Jesus Rodriguez-Manzano[2,3,11], Azumah Abdul-Tawab[12], Jamal-Deen Abdulai[13], Mohammed Abumanga[14], Jane Achan[15], Darlington A. Akogo[16], Angelina A. Amengu[17], Linda E. Amoah[18], Tochukwu D. Anyaduba[2], Emilia A. Udofia[19], Gordon A. Awandare[5,6], Frances B. da-Costa Vroom[20], Leonard Baatiema[21], Kevin Baker[3,15,22], Fatou Baldeh[1,23], Julie Balen[1,23], Jake Baum[24], Weston Baxter[17], Craig Bonnington[15], Salome A. Bukachi[25,26], Céire E. Costelloe[27,28], Luc P. de Witte[29], Samuel Duodu[5,6], Francis Dzabeng[5,13], Fahad A. Elnour[14], Joe Fitchett[30], Sahar Gamil[31], Sebastian S. Hachizovu[32], Prudence Hamade[15], Muzamil M. Abdel Hamid[33], Jethro A. Herberg[9,10], Waleed M. A. Jebreel[33], Jean-Bertin Kabuya[32], Flavia K. Bawa[5,6], Myrsini Kaforou[9,10], Dennis O. Laryea[12], Michael Levin[9,10], Christine Manyando[32], Abdelrahim O. Mohamed[31], Shola K. Molemodile Dele-Olowu[21], Nicolas Moser[2,3,8], Sydney Mwanza[34], Erick Odoyo[35], Patience Ofosuhemaa[36], Lucy C. Okell[37], Abubakr Omer[31], Francesca M. Piffer[9], Talya Porat[17], Anthony W. Sifuna[38], Maria Suau Sans[15], Faiza U. Bawah[13,39], Effua Usuf[1], Alfred E. Yawson[40], Shunmay Yeung[41] & Asadu Sserwanga[15]

[12]Disease Surveillance Department, Ghana Health Service, Accra, Ghana. [13]Department of Computer Science, University of Ghana, Accra, Ghana. [14]Patients Helping Fund, Khartoum, Sudan. [15]Malaria Consortium, London, UK. [16]minoHealth AI Labs, Accra, Ghana. [17]Dyson School of Design Engineering, Imperial College London, London, UK. [18]Immunology Department, Noguchi Memorial Institute for Medical Research, University of Ghana, Accra, Ghana. [19]Department of Community Health, University of Ghana Medical School, College of Health Sciences, University of Ghana, Accra, Ghana. [20]Department of Biostatistics, School of Public Health, College of Health Sciences, University of Ghana, Accra, Ghana. [21]Department of Health Policy, Planning and Management, School of Public Health, University of Ghana, Accra, Ghana. [22]Department of Global Public Health, Karolinska Institute, Stockholm, Sweden. [23]School of Nursing, Midwifery, Allied and Public Health, Canterbury Christ Church University, Canterbury, UK. [24]The University of New South Wales, Sydney, Australia. [25]Institute of Anthropology, Gender and African Studies, University of Nairobi, Nairobi, Kenya. [26]Department of Anthropology, Durham University, Durham, UK. [27]Health Informatics, Division of Clinical Studies, Institute of Cancer Research, London, UK. [28]Department of Primary Care and Public Health, School of Public Health,

Imperial College London, London, UK. [29]The Hague University of Applied Sciences, The Hague, Netherlands. [30]Institut Pasteur de Dakar, Dakar, Senegal. [31]Department of Biochemistry, Faculty of Medicine, University of Khartoum, Khartoum, Sudan. [32]Department of Public Health, National Health Research and Training Institute, Ndola, Zambia. [33]Institute of Endemic Diseases, University of Khartoum, Khartoum, Sudan. [34]Department of Biomedical Sciences, National Health Research and Training Institute, Ndola, Zambia. [35]Department of Biological Sciences, Masinde Muliro University of Science and Technology, Kakamega, Kenya. [36]Health Promotion Division, Ghana Health Service, Accra, Ghana. [37]MRC Centre for Global Infectious Disease Analysis, Department of Infectious Disease Epidemiology, Imperial College London, London, UK. [38]Department of Medical Biochemistry, Masinde Muliro University of Science and Technology, Kakamega, Kenya. [39]Department of Computer Science and Informatics, University of Energy and Natural Resources, Sunyani, Ghana. [40]Office of the Provost, College of Health Sciences, University of Ghana, Accra, Ghana. [41]Clinical Research Department, Faculty of Infections and Tropical Disease, London School of Hygiene and Tropical Medicine, London, UK.

