## [Transparent Peer Review file · Nature Communications]

Sensitive Near Point-of-Care Detection of Asymptomatic and Sub-microscopic Plasmodium falciparum Infections in African Endemic Countries

Corresponding Author: Dr Jesus Rodriguez-Manzano

Version 0:

Reviewer comments:

Reviewer #1

(Remarks to the Author)

General: Robinson et al. report on a LAMP-based molecular diagnostic platform that they adapted to Plasmodium molecular detection. The assay achieved modest analytical sensitivity. The assay was then used to test samples from persons with clinical disease and with asymptomatic infections. The authors suggest that the test has ideal performance characteristics to be used as a scalable and sensitive test that meets current WHO-defined field needs but my judgement is that the assay performance is somewhat underwhelming and hard to scale. While the paper has some interesting aspects, my judgement is that the data and the presentation of the approach is not sufficiently impactful or developed for publication in this journal.

Major:

1) Limited impact: The team adapted existing molecular assay technologies to the malaria question. The SmartLids system provides a neat approach for simplifying nucleic acid extraction. However, this is not Nat Comms material. This paper would be better suited to a more specialized molecular diagnostic journal where addition and some missing methodological info could be added. Much more work would be needed for publication in Nat Comms (including solving or addressing some of the limitations in sections 298-312). An interesting study for example could involve using the assay to answer a fundamental field epidemiology question that could not be answered by other available POC assays or by slower or more expensive molecular diagnostic assays. Since the analytical sensitivity of the Dragonfly/SmartLid assay is only "slightly better than expert microscopy" (line 227) and since ultrasensitive RDTs have also been developed, the assay ultimately seems like much too much work. I recognize that the authors have used Table S1 to make a case for the applicability of the assay in field use. However, given that competing approaches such as RT-LAMP can achieve much more analytically sensitive outcomes using similar technologies, I think that the approach presented here is less impactful than stated in the manuscript.

2) Incomplete info about assay performance: There are too few samples included in herein to have fully evaluated the assay. For example, there were 21 qPCR-positive asymptomatic infections and only six of those were RDT-negative, microscopy-negative, but molecular assay-positive. The data does not describe standard performance characteristics (for guidance see PMID: 20610823). The authors should more thoroughly consider factors that can create non-specific results such as a maximal assay reaction time (lines 193-194). Further, LoD estimates depend on the serial dilutions of the 3D7 cultures, but it seems like the cultured material may not have been as synchronous as intended. For example, when the assay is able to detect samples down to 0.04 parasites/uL, this means that the culture likely contains non-ring parasites like trophozoites and schizonts that contain higher numbers of genes per parasite-infected cell as compared to a ring stage infected cell with the minimal set of parasite genes per infected cell. The authors should also define whether the cultures were diluted into whole blood or some other matrix for the LoD studies (also line 331-332). The cultured serially diluted samples should also have been thoroughly tested by qPCR across the range of relevant samples. Without many more positive patient samples, assay performance would be better judged on the basis of parasite density, rather than on groupings of clinical vs. asymptomatic persons. For example, it would be typical to ascertain the parasite density of the samples using paired comparator qPCR samples and bin those infections into >100 para/uL, 5-100 para/uL, 0.5-5 para/uL, and 0.02-0.5 para/uL before judging the DragonFly performance against these bins.

3) Issues with scaling: The authors suggest that the assay is easily scalable. However, the assay requires considerable hands-on time and manual processes. For example, the section describing the SmartLid transfers and sample identification

specifies manually hand writing identifiers on lids to avoid cap swaps. The manual readout of the reaction and the suggestion that the pH of the buffer can shift all of the color scale also seem to be labor intensive and prone to errors. As such, this does not seem to be scalable and one could expect a lot of inter-operator error in field-use conditions. However, this is not addressed with real-world data. Although there is a limitations section in the conclusions, much of the paper reads more like an advertisement with somewhat preconceived notions about the utility and performance of the test.

4) Comparator molecular assay: The comparator assay is inadequately described. In reality, a more sensitive comparator should have been used to serve as a much more sensitive molecular gold standard. This would give the authors a much better understanding of what % of low-density infections were actually captured by the Dragonfly/SmartLid assay. The authors cite an assay LoD of 0.01 parasites/uL for the comparator qPCR but this is wrong for the way the qPCR assay was used in this manuscript. The stated LoD is from the Hofmann 2015 paper (Ref 46) where the samples consisted of blood fractions of 50–150 μ L. In contrast, in this manuscript, the comparator qPCR was used to test <5 uL of blood (based on three x 3 mm DBS discs), effectively 10-30X less blood. As such, the qPCR is only slightly more sensitive than the DragonFly assay but would fail to detect lower-density infections that are being increasingly reported. While the comparator qPCR was more sensitive, more nuance could be added to describe the rationale for use of that specific qPCR. All in all, the comparator qPCR could have been used more effectively in paired testing of control and experimental materials to characterize the novel assay.

5) Description of assay gene target(s): Gene target names need to be in the primary paper, not in the supplementals. One could leave the actual nucleic acid sequences to the Supplementals.

6) Missing clearer data and description of LoD and blood volumes as the primary driver of assay sensitivity: The analytical sensitivity of all NAATs is limited by blood volume. This is especially true as gene or RNA copy number increases. The section in lines 117-137 would benefit from more info about the actual sample input volumes.

Minor:

7) Line 40 and others: There are many instances throughout the paper where the authors should state “Plasmodium-infected” instead of “malaria-infected”. Also line 86

8) Line 59: This section on background assays should also include NASBA and RT-PCR for the abundant 18S rRNA targets. It might be better to introduce the term “NAATs” (nucleic acid amplification tests) as an umbrella term. Similarly, could substitute NAATs for PCR on line 68.

9) Line 65: “high analytical sensitivity” instead of “high performance”

10) Line 83: “for detecting Plasmodium genes”

11) Line 84: missing a closed parenthesis “)”

12) Line 137: “custom” rather than “bespoke”

13) Line 146: POC should have been defined earlier.

14) Line 183: “dye” not “die”

15) Table 1: contains lots of cells that are missing a space before the parentheses

Reviewer #2

(Remarks to the Author)

This manuscript describes the use of a LAMP-based diagnostic platform to be employed as a point-of-care highly sensitive diagnostic of asymptomatic and submicroscopic Plasmodium infection in Africa.

This is an unmet need in the real-world strategies put forward to eliminate malaria. Ultrasensitive molecular methods such as polymerase chain reaction (PCR) are difficult to implement in low-complexity and low-resource laboratory settings. In this context, the authors should be congratulated to have adapted a platform (originally developed to diagnose respiratory and skin infections in low resource settings) which is based on a magnetic bead extraction for sample preparation and using a pan/Pf loop mediated isothermal amplification potentially overcoming problems with standard commercially available LAMP. The study is a “proof-of-concept” conducted in high-tech laboratory using previously collected blood specimens from both symptomatic for malaria and asymptomatic subjects from Burkina Faso and The Gambia. This point should be clearly reported (for instance, at the end of the Introduction).

Methodology is well described and detail provided are good enough to be reproduced

As reported in the results the limit of detection (3.4 parasites/ μ L) and the overall sensitivity and specificity of this adapted LAMP compared with dried blood spot qPCR (used as gold standard) are in line with previous studies. The test showed a higher sensitivity (85.7%) compared to expert microscopy and rapid diagnostic tests among asymptomatic cases and was able to detect 83% of submicroscopic infections missed by both diagnostic tests. However, the main limitation of this study is the small number of asymptomatic cases (N=21) and subjects with low-level parasitaemia (< 16 parasites/uL).

Page 2 line 83- “SmartLid sample preparation technology” add “using magnetic bead extraction”

Page 7 lines 211-213- “the blood specimens from symptomatic individuals (n=50) were distributed across all age categories with a slight majority of females (58.1%). The blood samples from asymptomatic individuals (n=331) were predominantly from children under 15 years of age with equal sex ratio (1:1). What appears from table 1 is exactly the contrary. Please correct it. Moreover, about the overall asymptomatic subjects (n=331) please indicate (in the text or in the table) how many from the Gambia and how many from Burkina Faso.

Page 8 lines 241-243- and figure 7: “All samples with a parasite density greater than 16 parasites/uL were detected by expert microscopists. The difference in detection among the three diagnostic techniques for sample with a parasite density above 16 parasites/ μ L were not statistically significant (p>0.05)”. However, for parasite density between 16-100/ μ L the Dragonfly performed as RTD and missed 16.7% of cases against microscopy and for parasites > 200/ μ L missed 1.8% cases against microscopy. Besides the fact that the numbers were too small to draw any definite information, how do you explain

these results?

Page 13 line 393- "All slides were independently read by two qualified microscopists"

Page 13 line 405- In this case it is not specified if the results reported in the manuscript about the Dragonfly Pan/Pf malaria test were obtained by a single operator or by more than one. In the latter case how was the agreement? Moreover, because the lecture (positive or negative) was based on a colorimetric reaction (pink or yellow) I wonder if there were any "intermediate" reaction and if yes how many times? How many times the need to repeat the process?

Discussion – This is generally well-written reporting the strengths (use of lyophilized isothermal colorimetric LAMP, multiple blood samples-12- to be extracted and amplified, short time). As far as the limit of the study, only samples with Plasmodium falciparum infection (the main Plasmodium in sub-Saharan Africa) were used. No data about other species (applicability outside Africa). A second point is about "comprehensive cost analysis" (page 10 line 301). I think the authors should try to make a cost-analysis based on their experience in developing the platform and on the used reagents.

Briefly mention other platforms used for ultrasensitive detection of Plasmodium spp (for instance Wei H et al. Rapid and ultrasensitive detection of Plasmodium spp. Parasites via the RPA-CRISPR/Cas12a Platform. ACS Infectious Diseases 2023; 9: 1534-45).

Reviewer #3

(Remarks to the Author)

This manuscript describes a novel approach to detection of malaria parasite DNA using a combination of established LAMP methodology, and a novel medium-throughput DNA extraction workflow using a recently developed SmartLid methodology for manipulating magnetic beads through multiple steps without requiring specialized equipment. The SmartLid method was previously developed for respiratory swab samples and is here adapted for use with blood, with customized implements to increase the throughput up to 12 samples processed at a time.

LAMP has often been proposed for use with tropical diseases such as malaria, with excellent characteristics for use in low resource settings. While literature is inconsistent about whether an extraction is absolutely necessary with whole blood samples, vs simple lysis protocols, this manuscript demonstrates an approach to simplify DNA extraction with relatively simple and inexpensive equipment, and as such promises to make the LAMP technique both more reliable and more accessible. The authors also test performance with a sizable set of clinical samples, and demonstrate very good performance on par with gold standard qPCR. While the manuscript is well-written and the technology adaptation is clever, we are not sure it meets the expectation for Nature Communication in that it's not really a new scientific discovery, but rather focused on technology demonstration, with a strong clinical focus. Perhaps this article would be better suited for a clinical-focused journal like Lancet, Science Translational Medicine, or JAMA Network Open.

Minor comments/questions to address:

- There are a few typos/missing words (e.g., use 'concentrations' rather than 'densities' in line 60, move 'are commercially available' to line 76 after 'platforms', 'die' vs 'dye' in line 183, etc.)
- Do the 2 commercial LAMP tests achieve the sensitivity needed to detect asymptomatic individuals (less than 50-100 parasites/uL)?
- Suggestion to re-phrase the last sentence in the intro because the numbers don't add up and may cause confusion. "The samples included 331 asymptomatic and 50 symptomatic individuals. Ultimately, 71 malaria-positive cases were identified by DBS-qPCR."
- Why did you use a different RDT for 16 samples?
- How long are the lyophilized reagents stable and at what temps and humidity levels?
- What liquid are the positive and negative controls rehydrated with? Water? Buffer?
- How is the 24-tube block set up for both Pan and Pf-specific assays? Are the Pan and Pf primers in the same tube? Are there 2 strips with Pan primers and 2 strips with Pf primers (assays in separate tubes)?
- What genes are you targeting for your LAMP assays? This should be included in the methods.
- What happens when the color is orange (e.g., amplification has started but has not completed in 40 min)? Did users interpret 'in between' colors differently?
- Was the assay optimized (temperature, run time, primer concentration, etc.) in a previous study? If so, you should cite it.
- Both reviewers are confused by Figure 4A. Please explain or consider a different graphical format.
- Please write out confidence intervals for probit LOD analysis.
 - o If the assay occasionally detected 0.04 parasites/uL, how do you know those aren't false positives? You should test many more than 10 negative controls to establish a baseline for false positive rate with reasonable confidence intervals, to understand whether there is any likelihood that samples with very low parasite load might just occasionally exhibit spontaneous amplification (to which LAMP is sometimes prone).
- Was analytical specificity tested (e.g., evaluating cross-reactivity with near neighbors)?
- Statements in lines 211-213 don't match the data in Table 1. It seems like asymptomatic and symptomatic were switched.
- Figure 5: why not create full 2x2 contingency tables – one for symptomatic and one for asymptomatic individuals? This representation is a bit odd, and the reader has to calculate FN and FP rather than being stated. Equations for sensitivity and specificity from TP, TN, FP, FN should be stated in methods or at least referenced.
 - o Are these results for the Pan assay or for Pf or both? Is the DBS-qPCR a Pan or Pf-specific assay?
 - o There's no discussion on the Pan vs Pf-specific results? Were all samples that were positive for Pf also positive in the Pan assay?
- Figure 6 is also confusing because it's only part of a 2x2 contingency table.
 - o Where is specificity calculations for asymptomatic individuals?
 - o Why not also show data for symptomatic individuals?

- Figure 7: add <16 parasites/uL under submicroscopic label on graph
- o It seems microscopy is quite good above 16 parasites/uL – why isn't this used for asymptomatic screening?
- o Is Fisher's Exact Test the most appropriate? What about Cohen's Kappa Statistic?

Reviewer #4

(Remarks to the Author)

Version 1:

Reviewer comments:

Reviewer #1

(Remarks to the Author)

The paper is substantially improved with much more data, more impact, and more perspective. Nice work by the authors to make thoughtful revisions. I have a few additional comments and suggestions:

- It may be useful to approximate the volume that is captured by the DBS-qPCR sample (3 discs of 3 mm diameter = ~9 uL blood) to add to the perspective on the impact of sample volume on LoD.
- It seems like you could write *P. falciparum* after the first instance of *Plasmodium falciparum* and save some space as the full name is written out a lot.
- Lines 70-72 should also include reverse transcription PCR methods (PMIDs: 40461818, 38185134, 31017084) and RT-LAMP method (PMIDs: 33508016, 37041454) for completeness.
- Supplemental Table 4: Use "x" signs for dimensions (not *) for consistency. Also use decimal points or commas consistently for weight depending on journal standards. Check the other spacing in this table as there's some typographical errors and sloppy inconsistencies
- Over time, I would suggest that you develop something better than hand labeling the tube tops as a tracking system. 2D barcodes abound, and cell phone-based or other QR code scanner apps could be applied. Boring but important parts like this sample identification/labeling consideration are important when talking about scaling. It's not necessary from my perspective to expand on this in the paper herein, but I share this as guidance for ongoing implementation and roll-out of this strategy.
- The following lines 377-379 are not quite right since there is no WB-qPCR for the 672 capillary samples in the Supplementary data ("Detailed results obtained using Dragonfly, Alethia®, DBS-qPCR, and WB-qPCR for each of the 50 positive controls from confirmed malaria patients, as well as the 672 capillary blood samples, are provided in the Supplementary Data."). I don't see any WB-qPCR in the Supplementary File so best to either add the data if available or remove/edit the comment. I'm surprised that the specificity of your assay didn't suffer against DBS-qPCR since your DF LoD is better than the DBS-qPCR.

Reviewer #2

(Remarks to the Author)

This is the second revision of the manuscript entitled "High-sensitivity near point-of-care detection of asymptomatic and sub-microscopic *Plasmodium* infections in African endemic countries"

The authors have deeply revised and much improved their original manuscript. They have strengthened their results by adding more tested samples (submicroscopic parasitaemia) and conducting new experiments with comparison of analytical performance of different methods (Figure 4). The latter were performed with the collaboration of researchers at the School of Hygiene & Tropical Medicine and the Imperial College of London with the obvious need to add two coauthors (Lindsay B Stewart and Pantelis Georgiou).

Point 8: I know very well that *P. falciparum* is the main *Plasmodium* species in sub-Saharan Africa, and you used only *P. falciparum* positive samples in the study. Accordingly, add *falciparum* in the title.

I think that the suggested revisions have been formally addressed, and I would like to congratulate the authors on their significant efforts to do it.

Although the figures 1-3 are explicative I wonder if a short video clip can be added to the supplementary materials (if feasible).

References: the name of microorganism should be in italic with the lower case of the second name.

Reviewer #3

(Remarks to the Author)

The authors have satisfactorily addressed most of our previous comments.

The authors collected more samples from asymptomatic individuals which allowed them to highlight the sensitivity difference between Dragonfly (95%) and expert microscopy (71%). qPCR results confirmed that the samples included a broad range of parasite concentrations, including many at sub-microscopic concentrations. This is important because detecting sub-microscopic levels of the parasite is one of the advantages of Dragonfly. The authors also tested a large number of negative controls to address and disprove a possible false positive issue. Altogether, these significant changes make the article more impactful.

A few final requests (numbers correspond to items in response to reviewers):

5. Add length of time and temps that lyophilized reagents are stable for in the discussion

7. If Pan & Pf primers are in same tube, how do you distinguish which target is present? Is that not important? Please explain the rationale (in the manuscript) for having both targets in the same tube and the inability to distinguish whether the positive is from *P. falciparum* or broad Pan detection. Since the Pan primers in principle should detect *P. falciparum*, what is the benefit of including the Pf-specific primers in addition to the Pan primers? Is it intended to improve sensitivity for *P. falciparum* since this is the most prevalent species in the region where the study was conducted? And would a different strategy be appropriate (e.g. Pan only, or a different mixture of primers) in regions where a different species is more prevalent?

9. Please include text in the methods that even an “intermediate color shift (from the color reference control)” will be considered a positive test result.

23. Please add discussion as to why microscopy isn't practical – not scalable, time-consuming, personnel constraints. There were some great points in the response to reviewers, but no text was added to the manuscript.

Reviewer #4

(Remarks to the Author)

Reviewer #1

GENERAL: Robinson et al. report on a LAMP-based molecular diagnostic platform that they adapted to Plasmodium molecular detection. The assay achieved modest analytical sensitivity. The assay was then used to test samples from persons with clinical disease and with asymptomatic infections. The authors suggest that the test has ideal performance characteristics to be used as a scalable and sensitive test that meets current WHO-defined field needs but my judgement is that the assay performance is somewhat underwhelming and hard to scale. While the paper has some interesting aspects, my judgement is that the data and the presentation of the approach is not sufficiently impactful or developed for publication in this journal.

We appreciate the concerns raised by Reviewer #1 regarding the assay's analytical sensitivity and scalability/usability, and we welcome the opportunity to clarify and expand on these points. The manuscript has been extensively revised for clarity, and we have conducted multiple additional experiments to strengthen our findings and address Reviewer #1's concerns. These include: LOD comparison of our technology against DBS-qPCR, whole blood qPCR (WB-qPCR), and the Alethia® malaria platform, demonstrating a LOD of 0.6 parasites/μL for Dragonfly, an analytical specificity study with negative blood to better characterize the false positive rate at prolonged incubation times (N=120, incubated for 60min), as well as further evaluation of capillary blood samples collected at the community level, including an additional 341 samples of which 125 were positive. Please see answers below for further details. We believe these major study improvements, along with many minor updates detailed throughout the following responses, should satisfy Reviewer #1's primary objections to our manuscript.

The reviewer judges that “the assay performance is somewhat underwhelming and hard to scale”. We strongly disagree for the following reasons. First, the fundamental design principle of the Dragonfly/SmartLid system is scalability and ease of use in diverse field conditions. This is possible because Dragonfly uses **lyophilised reagents** that are stable at room temperature (2-30°C for 12+ months), obviating cold chain requirements. Next, it uses **portable isothermal heat blocks** that are inexpensive (112 GBP/unit) and can be powered by portable batteries. In terms of consumables, the entire workflow, with the exception of the custom molded SmartLid (and accompanying magnetic key), uses standard off-the-shelf components (e.g. 2 mL flip cap tubes, 8-tube PCR strips, etc), enabling easy and low-cost sourcing at scale all around the world. Finally, the workflow requires **minimal training** as it is designed for rapid training of local healthcare workers.

MAJOR comments:

1. Limited impact: The team adapted existing molecular assay technologies to the malaria question. The SmartLids system provides a neat approach for simplifying nucleic acid extraction. However, this is not Nat Comms material. This paper would be better suited to a more specialized molecular diagnostic journal where addition and some missing methodological info could be added. Much more work would be needed for publication in Nat Comms (including solving or addressing some of the limitations in sections 298-312). An interesting study for example could involve using the assay to answer a fundamental field epidemiology question that could not be answered by other available POC assays or by slower or more expensive molecular diagnostic assays. Since the analytical sensitivity of the Dragonfly/SmartLid assay is only “slightly better than expert microscopy” (line 227) and since ultrasensitive RDTs have also been developed, the assay ultimately seems like much too much work. I recognize that the authors have used Table S1 to make a case for the applicability of the assay in field use. However, given that competing approaches such as RT-LAMP can achieve much more analytically sensitive outcomes using similar technologies, I think that the approach presented here is less impactful than stated in the manuscript.

We appreciate the reviewer's assessment and diligent feedback regarding the impact and scope of our manuscript. We understand the high bar for publication in *Nature Communications* and value the detailed critique, which has guided our revisions.

We disagree, however, with the reviewer about the limited impact of the presented solution. Possibly, it was not clear from the manuscript that the main objective of developing a test such as Dragonfly malaria is to have a user-friendly test deployable at the most peripheral level that identifies as many malaria-infected asymptomatic carriers as possible.

The “slightly better than expert microscopy” sensitivity in the original submission refers to the analysis of a limited number of DBS-qPCR positive samples that included 50 symptomatic patients and only 21 asymptomatic

individuals. This means that most of these samples were from individuals with patent and relatively high parasitaemia and thus it is not surprising that Dragonfly and expert microscopy had similar sensitivity. In the original submission, when looking at the sensitivity for samples from asymptomatic individuals, the difference between Dragonfly and expert microscopy is more marked although not statistically significant due to the small sample size and thus lack of statistical power. This was a limitation that we have addressed in the new submission by increasing the sample size of asymptomatic carriers. Indeed, the sensitivity analysis in the updated version is performed on a total of 196 qPCR-confirmed positive samples from individuals recruited at community level. The updated results show a significantly higher sensitivity of Dragonfly (95.2%) compared to expert microscopy (70.5%).

As described in the "Validation of the Dragonfly platform against RDT and light microscopy using DBS-qPCR as the gold standard" section (Lines 234-243): "First, a total of 50 capillary blood specimens from febrile malaria patients, purposively selected as positive controls based on concordant Plasmodium detection by LM, RDT, and DBS-qPCR, were also tested using the Dragonfly platform. All 50 samples tested positive, confirming the compatibility of our method with real, field-collected capillary blood samples prior to evaluation on unlabelled community survey samples. Next, 672 capillary blood samples collected at the community-level in The Gambia and Burkina Faso were used to evaluate the performance of the Dragonfly platform against DBS-qPCR as the reference method. All samples were also assessed using expert LM and RDTs to benchmark the performance of our method against the two standard diagnostic approaches for malaria. Of the 672 samples, 27.1% (146/672) were positive for *P. falciparum* by DBS-qPCR. These positive samples represented a broad range of parasite densities, including both microscopically detectable (n=103) and submicroscopic infections (n=43)." See the updated Supplementary data and Figure 6 for further details.

Regarding the two LAMP-based technologies that have been commercialized to date, we compared Dragonfly and Alethia. The two tests gave similar results. However, despite being available on the market, Alethia is mainly used in high-income settings, likely because of the high cost of both the required instrument (16,000 GBP/unit) and the individual test (43 GBP). As for the ultrasensitive RDTs, a meta-analysis by Yimam *et al.* (2022) reported a pooled sensitivity of only 50% (95% CI: 33–68%), highlighting the limitations in detecting low-density infections.

Lastly, we would like to push back on the reviewer's statement that the diagnostic test is "not sufficiently impactful or developed for Nat Comms" / "better suited to a more specialized molecular diagnostic journal". First, our work is not merely about the adaptation of an existing assay; rather, it presents the development and realistic initial validation of a system designed to bridge the gap between laboratory-grade molecular sensitivity and point-of-care implementation for challenging environments where cost and portability are paramount. This transformative potential for surveillance and targeted interventions in global health, particularly in malaria-endemic regions, aligns with the scope of *Nature Communications* for publishing advances that provide "a significant step forward in a particular field of research." Our work addresses a critical epidemiological challenge (i.e., identifying the hidden reservoir of infection) with a deployable technological solution.

2. Incomplete info about assay performance: There are too few samples included in herein to have fully evaluated the assay. For example, there were 21 qPCR-positive asymptomatic infections and only six of those were RDT-negative, microscopy-negative, but molecular assay-positive. The data does not describe standard performance characteristics (for guidance see PMID: 20610823).

As mentioned above, we agree that the number of qPCR-positive asymptomatic infections in the original submission was small, and this was a major limitation. To address this, we have now included in the analysis a total of 672 samples collected at community level. Of these, 146 were qPCR positives, with 43 of them with a parasite density below the detection limit of expert light microscopy. A summary of the sample breakdown by category is shown in Figure 5 of the main manuscript.

3. The authors should more thoroughly consider factors that can create non-specific results such as a maximal assay reaction time (lines 193-194).

To address this concern and provide robust evidence for assay specificity, we have now included a study evaluating 120 whole blood negative samples using the Dragonfly malaria test at incubation times of up to 60 minutes. This experiment yielded a specificity of 98.3% (95% CI: 96.0–100). These additional data reinforce the high specificity of the assay across the tested reaction times and confirm that the recommended maximum incubation time for the standard test panel is 40 minutes, which was used for all analytical and clinical specimens

referenced in the manuscript. This information has now been incorporated into the Results section of the main manuscript, Supplementary Table 2, and the Supplementary Data file (see the “specificity_DF_data” sheet).

It is important to highlight that the study now includes a total of 526 negative specimens (capillary blood), characterised by DBS-qPCR, which represent 216 additional negative samples. As described in Figure 6, this represents a specificity of 96.8 (95% CI: 94.9-98) for the Dragonfly malaria test.

We also evaluated the contamination risk of the SmartLid workflow as part of another work. Alternating high-positive samples (spiked with 200uL of 1×10^6 pfu/mL of SARS-CoV-2 viral particles) and negative control samples were extracted in close proximity in a single batch. As shown by the raw RT-qPCR readout (Lightcycler® LC96, Roche) below, all positive samples amplified while all negative samples did not amplify. These results, in combination with anecdotal evidence across many studies similar to the one presented in this manuscript, demonstrate the low cross-contamination risk of the SmartLid extraction process, even when using high-throughput workflows for 12 simultaneous extractions.

4. Further, LoD estimates depend on the serial dilutions of the 3D7 cultures, but it seems like the cultured material may not have been as synchronous as intended. For example, when the assay is able to detect samples down to 0.04 parasites/uL, this means that the culture likely contains non-ring parasites like trophozoites and schizonts that contain higher numbers of genes per parasite-infected cell as compared to a ring stage infected cell with the minimal set of parasite genes per infected cell. The authors should also define whether the cultures were diluted into whole blood or some other matrix for the LoD studies (also line 331-332). The cultured serially diluted samples should also have been thoroughly tested by qPCR across the range of relevant samples.

We confirm that for the LoD experiments, the 3D7 cultures were diluted into whole blood. This information has now been integrated into the Results and Methods sections of the manuscript as correctly suggested. Furthermore, in agreement with the reviewer's suggestion for a more robust LoD assessment, we have conducted an additional and extensive comparison of analytical performance in collaboration with the London School of Hygiene & Tropical Medicine (LSHTM). This validation utilised EDTA-blood spiked with ring-stage *P. falciparum* 3D7 strain, tested across four molecular-based methods: Dragonfly, Alethia, DBS-qPCR, and whole-blood qPCR. To address the reviewer's concern regarding the potential presence of non-ring stage parasites, we employed a validated magnetic purification procedure to ensure that the final preparation consisted essentially of ring-stage parasites (<https://malariajournal.biomedcentral.com/articles/10.1186/1475-2875-7-45>). A representative image of the thin blood smear from the final preparation is now provided in Supplementary Figure 2. Serial dilutions were performed from 6,000 parasites/ μ L down to 0.125 parasites/ μ L, with lower concentrations tested in 10 replicates. This comprehensive approach provides a thorough characterization of our assay's analytical sensitivity and has now been integrated in Figure 4 of the manuscript.

5. Without many more positive patient samples, assay performance would be better judged on the basis of parasite density, rather than on groupings of clinical vs. asymptomatic persons. For example, it would be typical to ascertain the parasite density of the samples using paired comparator qPCR samples and bin those infections into >100 para/uL, 5-100 para/uL, 0.5-5 para/uL, and 0.02-0.5 para/uL before judging the DragonFly performance against these bins.

In line with the reviewers' suggestions, we restricted the diagnostic performance evaluation to samples collected at community level ($n=672$), primarily from asymptomatic individuals. As the qPCR assays were not performed using standards due to established SOPs in the partners laboratories, we were unable to quantify parasite densities by qPCR. Accordingly, we instead estimated the parasite density by microscopy. This allows determining the fraction of low-density *Plasmodium* infections that would have been missed by conventional

microscopy and/or rapid diagnostic tests (RDTs) but correctly identified by the Dragonfly assay. In addition, the analytical sensitivity experiments now provide a complementary evaluation of Dragonfly performance across defined parasite-density level.

6. Issues with scaling: The authors suggest that the assay is easily scalable. However, the assay requires considerable hands-on time and manual processes. For example, the section describing the SmartLid transfers and sample identification specifies manually handwriting identifiers on lids to avoid cap swaps. The manual readout of the reaction and the suggestion that the pH of the buffer can shift all of the color scale also seem to be labor intensive and prone to errors. As such, this does not seem to be scalable, and one could expect a lot of inter-operator error in field-use conditions. However, this is not addressed with real-world data. Although there is a limitations section in the conclusions, much of the paper reads more like an advertisement with somewhat preconceived notions about the utility and performance of the test.

While we acknowledge that the test does not reach the level of ease of use offered by RDT or expensive automated systems, the experience of the field team implementing the test in The Gambia and Burkina Faso suggests a much more positive outlook than provided by the reviewer's comment above. This will be documented in a qualitative work conducted by the social scientists on the usability and feasibility of the test in resource constrained-settings (*data analysis in progress*). Furthermore, we recently invited an implementation scientist from the NGO Malaria Consortium (<https://www.malariaconsortium.org/>) to observe Dragonfly in operation and his opinion was that the test was suitable for large scale screening. Therefore, we can assure that the SmartLid transfer, and the manual identification are, in practice, less arduous than they appear. The design allows for a relatively quick and intuitive workflow, and the manual handwriting of identifiers on the lids is a safe step for robust sample tracking. Please note that also the Alethia system requires meticulous manual labelling during sample preparation and loading into the instrument to maintain sample integrity and traceability.

7. Comparator molecular assay: The comparator assay is inadequately described. In reality, a more sensitive comparator should have been used to serve as a much more sensitive molecular gold standard. This would give the authors a much better understanding of what % of low-density infections were actually captured by the Dragonfly/SmartLid assay. The authors cite an assay LoD of 0.01 parasites/ μ L for the comparator qPCR but this is wrong for the way the qPCR assay was used in this manuscript. The stated LoD is from the Hofmann 2015 paper (Ref 46) where the samples consisted of blood fractions of 50–150 μ L. In contrast, in this manuscript, the comparator qPCR was used to test <5 μ L of blood (based on three x 3 mm DBS discs), effectively 10-30X less blood. As such, the qPCR is only slightly more sensitive than the DragonFly assay but would fail to detect lower-density infections that are being increasingly reported. While the comparator qPCR was more sensitive, more nuance could be added to describe the rationale for use of that specific qPCR. All in all, the comparator qPCR could have been used more effectively in paired testing of control and experimental materials to characterize the novel assay.

We thank the reviewer for highlighting the error regarding the LOD of the DBS-qPCR. Indeed, the value of 0.01 parasites/ μ L is applicable to assays using a higher blood volume. We have revised the manuscript accordingly. Although some lower-density infections may have been missed by the PCR comparator, we believe our validation remains robust. Modelling studies have demonstrated that a diagnostic test capable of detecting parasite densities as low as 2 parasites/ μ L can significantly reduce malaria prevalence and incidence by test-and-treat strategies, by identifying up to 95% of the infectious individuals. Therefore, Dragonfly will identify a substantial proportion of malaria carriers at the level required to accelerate malaria elimination efforts.

8. Description of assay gene target(s): Gene target names need to be in the primary paper, not in the supplementals. One could leave the actual nucleic acid sequences to the Supplementals.

We have now specified the Pan-Plasmodium (Pan18s) and *P. falciparum* (PfK13, Pf mtDNA) assays' gene names in the methods section of the main manuscript, while their primer sequences can be found in Supplementary Table 1.

9. Missing clearer data and description of LoD and blood volumes as the primary driver of assay sensitivity: The analytical sensitivity of all NAATs is limited by blood volume. This is especially true as gene or RNA copy number increases. The section in lines 117-137 would benefit from more info about the actual sample input volumes.

We agree with the reviewer's comment, and we have now included the input details previously mentioned in the results section (in the new analytical performance section of the manuscript).

MINOR comments:

- a) Line 40 and others: There are many instances throughout the paper where the authors should state "Plasmodium-infected" instead of "malaria-infected". Also line 86
- b) Line 59: This section on background assays should also include NASBA and RT-PCR for the abundant 18S rRNA targets. It might be better to introduce the term "NAATs" (nucleic acid amplification tests) as an umbrella term. Similarly, could substitute NAATs for PCR on line 68.
- c) Line 65: "high analytical sensitivity" instead of "high performance"
- d) Line 83: "for detecting Plasmodium genes"
- e) Line 84: missing a closed parenthesis ")"
- f) Line 137: "custom" rather than "bespoke"
- g) Line 146: POC should have been defined earlier.
- h) Line 183: "dye" not "die"
- i) Table 1: contains lots of cells that are missing a space before the parentheses

We are grateful to the reviewer for the valuable and detailed suggestions. We have thoroughly reviewed the manuscript and implemented all proposed amendments, including the corrections for wording, typographical errors and formatting inconsistencies that are now highlighted in the manuscript.

Overall, we thank Reviewer #1 for their detailed, critical, and insightful review of our draft manuscript. Through addressing their points with the corrections above, including both the Major and Minor points, we feel the new draft is now significantly more robust.

Reviewer #2 (Remarks to the Author):

1. This manuscript describes the use of a LAMP-based diagnostic platform to be employed as a point-of-care highly sensitive diagnostic of asymptomatic and submicroscopic Plasmodium infection in Africa. This is an unmet need in the real-world strategies put forward to eliminate malaria. Ultrasensitive molecular methods such as polymerase chain reaction (PCR) are difficult to implement in low-complexity and low-resource laboratory settings. In this context, the authors should be congratulated to have adapted a platform (originally developed to diagnose respiratory and skin infections in low resource settings) which is based on a magnetic bead extraction for sample preparation and using a pan/Pf loop mediated isothermal amplification potentially overcoming problems with standard commercially available LAMP.

We thank the reviewer for their insightful and highly encouraging comments. We appreciate the reviewer's recognition of the critical unmet need our work addresses *i.e.*, the need for a highly sensitive, point-of-care diagnostic capable of detecting asymptomatic and submicroscopic *Plasmodium* infections in Africa.

2. The study is a "proof-of-concept" conducted in high-tech laboratory using previously collected blood specimens from both symptomatic for malaria and asymptomatic subjects from Burkina Faso and The Gambia. This point should be clearly reported (for instance, at the end of the Introduction).

We have now included this detail in the introduction section of the manuscript.

3. Methodology is well described and detail provided are good enough to be reproduced. However, the main limitation of this study is the small number of asymptomatic cases (N=21) and subjects with low-level parasitaemia (< 16 parasites/uL).

This concern has now been addressed, as detailed in our response to Reviewer #1. In brief, we have included an additional 341 samples, of which 146 were *P. falciparum* positive. Among these, 43 specimens had parasite densities below the detection limit of expert light microscopy. A summary of the sample breakdown by clinical category is presented in Figure 5 of the main manuscript.

4. Page 2 line 83- "SmartLid sample preparation technology" add "using magnetic bead extraction"

This has now been added in the manuscript.

5. Page 7 lines 211-213- “the blood specimens from symptomatic individuals (n=50) were distributed across all age categories with a slight majority of females (58.1%). The blood samples from asymptomatic individuals (n=331) were predominantly from children under 15 years of age with equal sex ratio (1:1). What appears from table 1 is exactly the contrary. Please correct it. Moreover, about the overall asymptomatic subjects (n=331) please indicate (in the text or in the table) how many from the Gambia and how many from Burkina Faso.

We thank the reviewer for meticulously pointing out these inconsistencies. The descriptive text in the manuscript related to Table 1 has been revised to reflect the updated data. Additionally, as suggested by the reviewer, the breakdown of the overall subjects by country has also been added to Table 1 for clarity.

6. Page 8 lines 241-243- and figure 7: “All samples with a parasite density greater than 16 parasites/uL were detected by expert microscopists. The difference in detection among the three diagnostic techniques for sample with a parasite density above 16 parasites/ μ L were not statistically significant ($p>0.05$)”. However, for parasite density between 16-100/ μ L the Dragonfly performed as RTD and missed 16.7% of cases against microscopy and for parasites $> 200/\mu$ L missed 1.8% cases against microscopy. Besides the fact that the numbers were too small to draw any definite information, how do you explain these results?

We appreciate the reviewer's valuable statistical observation. To address this, we have increased the total sample size and updated **Figure 7** accordingly, which now presents the detection rates of DBS-qPCR positive samples by RDT, microscopy, and Dragonfly, categorized by parasite density.

Notably, the Dragonfly system correctly identified **95.3% of all submicroscopic infections** (parasite density <16 parasites/ μ L), demonstrating a statistically significant difference ($p<0.05$). In contrast, for parasite densities above 200 parasites/ μ L, no statistically significant differences were observed between the three techniques ($p>0.05$). The missed infections by Dragonfly with parasites densities $\geq 16/\mu$ L, including a few at $>200/\mu$ L, may be due to either sample integrity at the time of testing or human error. However, these missed infections did not significantly impact the Dragonfly's overall performance.

7. Page 13 line 393- “All slides were independently read by two qualified microscopists” Page 13 line 405- In this case it is not specified if the results reported in the manuscript about the Dragonfly Pan/Pf malaria test were obtained by a single operator or by more than one. In the latter case how was the agreement? Moreover, because the lecture (positive or negative) was based on a colorimetric reaction (pink or yellow) I wonder if there were any “intermediate” reaction and if yes how many times? How many times the need to repeat the process?

We thank the reviewer for raising this point regarding the interpretation of colorimetric results. In our colorimetric LAMP assay, the colour change reflects a shift in pH due to the release of protons during DNA amplification and anything that does not correspond to the initial colour control (pink) is classified as an amplification product. This results in a clear and distinct colour change. However, in the extremely rare case an intermediate colour appears, which may be due to subtle levels of inhibitory substances partially impeding the completed reaction in the recommended 40min incubation time, this should still be (and was for the purpose of the study) considered a positive outcome. As described in the “Multi-patient malarial Pan/Pf test panel design” section of the Results (Line 183), this was the exact purpose of including the “colour reference control.” As stated in the manuscript:

Lines 193-199: “First, as the reaction utilises an unbuffered LAMP system and pH sensitive dye to detect amplification, a colour reference control reaction was included, which omits polymerase enzymes to prevent amplification. This reaction will always remain pink with the exact shade of pink varying based on the starting pH of the eluted nucleic acids. Depending on the sample type (e.g. respiratory swab eluent versus whole blood), this starting pH and subsequent colour can vary slightly. Therefore, this reaction provided a reference colour against which all other reaction outcomes were compared.”

All operators received comprehensive training and results reported in the manuscript for the Dragonfly were obtained by multiple trained operators. Therefore, while multiple individuals performed the tests, the standardised protocol and objective readout reduce the potential for significant inter-operator disagreement. We did not observe any discrepancies attributed to operator variability during the study.

8. Discussion – This is generally well-written reporting the strengths (use of lyophilized isothermal colorimetric LAMP, multiple blood samples-12- to be extracted and amplified, short time). As far as the limit of the study, only samples with Plasmodium falciparum infection (the main Plasmodium in sub-Saharan Africa) were used. No data

about other species (applicability outside Africa). A second point is about “comprehensive cost analysis” (page 10 line 301). I think the authors should try to make a cost-analysis based on their experience in developing the platform and on the used reagents.

We appreciate the reviewer's positive feedback on the clarity and strengths of our report.

Regarding the limitations highlighted, we acknowledge the focus on *P. falciparum* due to its epidemiological predominance in sub-Saharan Africa, our primary target region. While the current study specifically evaluated our platform against *P. falciparum* infections, the underlying LAMP technology is broadly applicable to other Plasmodium species. Future studies will aim to include a broader range of Plasmodium species to demonstrate the assay's applicability outside of *P. falciparum*-endemic regions and its potential as a pan- Plasmodium diagnostic tool.

Concerning the "comprehensive cost analysis", we agree about the importance of including this, even at the prototype stage, and have thus now added a preliminary cost analysis as Supplementary Table 5.

9. Briefly mention other platforms used for ultrasensitive detection of Plasmodium spp (for instance Wei H et al. Rapid and ultrasensitive detection of Plasmodium spp. Parasites via the RPA-CRISPR/Cas12a Platform. ACS Infectious Diseases 2023; 9: 1534-45)

This has now been added in the Introduction section (Line 71-72).

Reviewer #3 (Remarks to the Author)

This manuscript describes a novel approach to detection of malaria parasite DNA using a combination of established LAMP methodology, and a novel medium-throughput DNA extraction workflow using a recently developed SmartLid methodology for manipulating magnetic beads through multiple steps without requiring specialized equipment. The SmartLid method was previously developed for respiratory swab samples and is here adapted for use with blood, with customized implements to increase the throughput up to 12 samples processed at a time.

LAMP has often been proposed for use with tropical diseases such as malaria, with excellent characteristics for use in low resource settings. While literature is inconsistent about whether an extraction is absolutely necessary with whole blood samples, vs simple lysis protocols, this manuscript demonstrates an approach to simplify DNA extraction with relatively simple and inexpensive equipment, and as such promises to make the LAMP technique both more reliable and more accessible. The authors also test performance with a sizable set of clinical samples, and demonstrate very good performance on par with gold standard qPCR.

While the manuscript is well-written and the technology adaptation is clever, we are not sure it meets the expectation for Nature Communication in that it's not really a new scientific discovery, but rather focused on technology demonstration, with a strong clinical focus. Perhaps this article would be better suited for a clinical-focused journal like Lancet, Science Translational Medicine, or JAMA Network Open.

We thank Reviewer #3 for their thoughtful feedback and for acknowledging the quality of the manuscript and the technological innovation presented. Regarding the suitability of this work for Nature Communications, we respectfully refer to our response to Reviewer #1, where we provide a detailed justification.

MINOR comments:

1. There are a few typos/missing words (e.g., use ‘concentrations’ rather than ‘densities’ in line 60, move ‘are commercially available’ to line 76 after ‘platforms’, ‘die’ vs ‘dye’ in line 183, etc.)

We thank the reviewer for the suggestions. We have reviewed the manuscript and addressed the proposed corrections that are now highlighted in the manuscript.

2. Do the 2 commercial LAMP tests achieve the sensitivity needed to detect asymptomatic individuals (less than 50-100 parasites/uL)?

We thank the reviewer for the pertinent question. According to Eiken's Loopamp™ Malaria Detection (Eiken Chemical Co., Tokyo, Japan) materials and studies and based on our analysis of the commercial Alethia Malaria (Meridian Bioscience Inc., Cincinnati, OH, USA), both platforms possess the requisite analytical sensitivity needed to detect *Plasmodium falciparum* infections in asymptomatic individuals. Specifically, the Loopamp

Malaria Pf Detection Kit is reported to have a LoD of approximately 1-2 parasites/uL. The Alethia Malaria system as shown in Figure 4 and Supplementary Table 3, demonstrated an even lower LoD (< 1 parasite/uL) for *P. falciparum*. Given that asymptomatic infections are frequently characterized by parasite densities below 50–100 parasites/μL the LoDs of these commercial LAMP platforms directly address the sensitivity requirement for identifying sub-microscopic and low-density infections. However, their use at large scale in African settings are constrained by the high cost and limited throughput, particularly in the case of Alethia Malaria (with an instrument cost of £16,000 and a test cost of > £40).

3. Suggestion to re-phrase the last sentence in the intro because the numbers don't add up and may cause confusion. "The samples included 331 asymptomatic and 50 symptomatic individuals. Ultimately, 71 malaria-positive cases were identified by DBS-qPCR."

We agree with the reviewer, the sentence has been updated to reflect the new data and in accordance with the reviewer's suggestion to improve clarity.

4. Why did you use a different RDT for 16 samples?

The reason for this variation was purely logistical, stemming from the fact that the samples were collected from different health facilities using different RDT brands. It is important to emphasize that both RDTs used were based on HRP2- detection and, critically, were WHO-approved *in vitro* diagnostic tests. This ensured that, despite the brand difference, the underlying detection mechanism and diagnostic performance were comparable and met international standards for malaria diagnosis in symptomatic cases.

5. How long are the lyophilized reagents stable and at what temps and humidity levels?

The Lyophilised reagents for the Dragonfly system are stable at room temperature for at least 6 months, with a range of temperature going from -20 to +30°C. The use of hermetic packaging containing desiccant ensures protection from moisture and allow the test panel reagents to be stable from the date of manufacture, while the SmartLid extraction kits have a shelf-life of more than 12 months. When stored in the fridge, however, both components can achieve significantly longer shelf lives. We are currently exploring new suppliers and manufacturing processes to extend the shelf life of the lyophilised reagents.

6. What liquid are the positive and negative controls rehydrated with? Water? Buffer?

Both the positive extraction controls (EDTA blood spiked with ring-stage *Plasmodium falciparum* 3D7) and negative extraction controls (EDTA blood from a healthy donor) were rehydrated using the provided Dragonfly elution buffer, which consists primarily of molecular-grade water.

7. How is the 24-tube block set up for both Pan and Pf-specific assays? Are the Pan and Pf primers in the same tube? Are there 2 strips with Pan primers and 2 strips with Pf primers (assays in separate tubes)?

Thank you for the question regarding the configuration of the 24-tube block for the combined *Pan* and *Pf*-specific assays. This clarification is indeed important for understanding the assay's practical implementation and throughput. To elaborate, the design of the assay tubes within the 24-tube block is configured as follows:

- The Pan/Pf primers are co-lyophilised within the same reaction tube meaning that a single reaction well simultaneously amplifies targets for Plasmodium genus (Pan) and *Plasmodium falciparum* (Pf), allowing for a comprehensive detection approach in a single reaction.
- Each 8-tube strip is precisely arranged to include: 6 identical wells containing the Pan/Pf LAMP assay reagents for patient samples and 2 dedicated control wells (a colour control and an internal control) which are essential for monitoring and validating the assay run and ensuring the integrity of the extraction and amplification process.
- Given that each heat block unit can accommodate a total of 4 strips of 8 tubes, this allows for the simultaneous incubation of up to 24 total samples per isothermal heat block.

We have since clarified this in the manuscript to reflect that 12 samples are able to be extracted at time, and thus leading to two strips of 6 Pan/pf reactions each amplifying the 12 samples simultaneously as well. The reference to 24 has been removed.

8. What genes are you targeting for your LAMP assays? This should be included in the methods.

We have now specified the Pan-Plasmodium (Pan18s) and *P.falciparum* (PfK13, Pf mtDNA) assay's gene names in the main manuscript, while the primer sequences can be found in Supplementary Table 3.

9. What happens when the color is orange (e.g., amplification has started but has not completed in 40 min)? Did users interpret 'in between' colors differently?

Thank you for this query. This point has been addressed in our response to Reviewer 2. To reiterate: In our colorimetric LAMP assay, the colour change reflects a shift in pH due to the release of protons during DNA amplification and anything that does not correspond to the initial colour control (pink) is classified as an amplification product. This results in a clear and distinct colour change. However, in the extremely rare case an intermediate colour appears, which may be due to subtle levels of inhibitory substances partially impeding the reaction, this should still be (and was for the purpose of the study) considered a positive outcome. As described in the "Multi-patient malarial Pan/Pf test panel design" section of the Results (Line 183), this was the exact purpose of including the "colour reference control." As stated in the manuscript (Lines 193-199): "*First, as the reaction utilises an unbuffered LAMP system and pH sensitive dye to detect amplification, a colour reference control reaction was included, which omits polymerase enzymes to prevent amplification. This reaction will always remain pink with the exact shade of pink varying based on the starting pH of the eluted nucleic acids. Depending on the sample type (e.g. capillary blood from finger pricks versus dried blood spot eluates), this starting pH and subsequent colour can vary slightly. Therefore, this reaction provided a reference colour against which all other reaction outcomes were compared.*"

All operators received comprehensive training and results reported in the manuscript for the Dragonfly were obtained by multiple trained operators. Therefore, while multiple individuals performed the tests, the standardised protocol and objective readout reduce the potential for significant inter-operator disagreement. We did not observe any discrepancies attributed to operator variability during the study.

10. Was the assay optimized (temperature, run time, primer concentration, etc.) in a previous study? If so, you should cite it.

Yes, that is correct, we did not perform *de novo* optimization of these parameters but rather assessed the performance of the pre-optimised systems in a relevant clinical or research context. The LAMP malaria assays developed in our previous study by our team, Malpartida-Cardenas *et al*, 2023 (<https://pubmed.ncbi.nlm.nih.gov/37158750/>), has now been cited in the main manuscript.

11. Both reviewers are confused by Figure 4A. Please explain or consider a different graphical format.

Thank you to the reviewers for highlighting the ambiguity in the original Figure 4A. We agree that its graphical format led to confusion and hindered the clear interpretation of the data. To address this, we have revised the presentation of this information, and the new data are now presented with a different graphical format to enhance the clarity of the analytical sensitivity study.

12. Please write out confidence intervals for probit LOD analysis.

This has now been added in the results section of the manuscript.

13. If the assay occasionally detected 0.04 parasites/uL, how do you know those aren't false positives? You should test many more than 10 negative controls to establish a baseline for false positive rate with reasonable confidence intervals, to understand whether there is any likelihood that samples with very low parasite load might just occasionally exhibit spontaneous amplification (to which LAMP is sometimes prone).

We agree with the reviewer's comment that more negative controls were required. This was also raised by Reviewer #1. We have now addressed this concern by significantly increasing the number of negative samples tested at different reaction times. This information has now been incorporated in the Results section of the manuscript (Multi-patient malaria Pan/Pf test panel design) as well as in the Supplementary Table 2.

14. Was analytical specificity tested (e.g., evaluating cross-reactivity with near neighbours)?

Analytical specificity was assessed during assay optimisation to ensure the selective detection of *Plasmodium falciparum* DNA while minimising cross-reactivity with closely related *Plasmodium* species. For further details on the empirical and *in silico* analyses, see Malpartida-Cardenas *et al.* 2023 (<https://pubmed.ncbi.nlm.nih.gov/37158750/>). The *Pf* LAMP primers used demonstrated no cross-reactivity with other *Plasmodium* species.

Please see response 3 to Reviewer 1 for further details on specificity testing.

15. Statements in lines 211-213 don't match the data in Table 1. It seems like asymptomatic and symptomatic were switched.

We thank the reviewer for their attention to detail and now have addressed this concern.

16. Figure 5: why not create full 2x2 contingency tables – one for symptomatic and one for asymptomatic individuals? This representation is a bit odd, and the reader has to calculate FN and FP rather than being stated. Equations for sensitivity and specificity from TP, TN, FP, FN should be stated in methods or at least referenced.

Thank you for your feedback regarding Figure 5. We have now improved the presentation of the diagnostic performance data comparing the clinical performance of the Dragonfly, RDT, and expert microscopy methods, in 2x2 contingency tables as suggested. This is now Figure 6 of the manuscript, as well as Supplementary Figure 1. Finally, equations have also been added as Supplementary Table 6 as suggested.

17. Are these results for the Pan assay or for Pf or both? Is the DBS-qPCR a Pan or Pf-specific assay?

The question related to the Dragonfly malaria Pan/Pf assay has been clarified in question number 7, Reviewer #3. Following the established standard operation procedures of partners countries where capillary specimens were collected, and since *P. falciparum* is the predominant species in the study areas, the DBS-qPCR used as comparator is specific to *P. falciparum*.

18. There's no discussion on the Pan vs Pf-specific results? Were all samples that were positive for Pf also positive in the Pan assay?

This point has been addressed in our response to Question 7 from Reviewer #3. In the 8-strip tube format, each of the six reactions used to test clinical specimens contained both pan-*Plasmodium* and *P. falciparum* primers within the same well—an approach known as tandem-LAMP. This strategy was employed to increase the probability of detecting specimens with submicroscopic parasite densities.

19. Figure 6 is also confusing because it's only part of a 2x2 contingency table.

Thank you for your feedback regarding Figure 6, we agree and have presented the diagnostic performance data in a new format as described above.

20. Where is specificity calculations for asymptomatic individuals?

This information has been added as Supplementary Figure 1, in the same format as manuscript Figure 6.

21. Why not also show data for symptomatic individuals?

We have now included the data for symptomatic individuals in Supplementary Figure 1 as well.

22. Figure 7: add <16 parasites/uL under submicroscopic label on graph

This label has now been added. We agree that it improves the clarity of the figure.

23. It seems microscopy is quite good above 16 parasites/uL – why isn't this used for asymptomatic screening?

Indeed, microscopy performs well at parasite densities above 16 parasites/ μL . However, practical limitations make it unsuitable for large-scale asymptomatic screening. The preparation and reading of a slide, especially at low parasite densities, can take an experienced microscopist at least 25–30 minutes per sample. In contrast, Dragonfly malaria can deliver results for up to 12 samples simultaneously in just 45 minutes from start to finish. By using multiple heat blocks per operator, the throughput can be scaled significantly, making it far more efficient than microscopy for mass screening.

Moreover, timely diagnosis is essential for enabling prompt treatment during active case detection campaigns. Achieving this with microscopy alone would be extremely challenging, if not unfeasible, due to time and personnel constraints.

It is also important to note that the microscopy results reported in our study were generated by expert microscopists, able of consistently detecting parasite densities in the range of 16–100 parasites/ μL . In real-world settings, non-expert microscopists typically have a detection threshold above 100 parasites/ μL , limiting the effectiveness of microscopy for detecting low-density asymptomatic infections.

24. Is Fisher’s Exact Test the most appropriate? What about Cohen’s Kappa Statistic?

We used McNemar’s test in the updated analysis to evaluate whether detection rates among PCR-positive samples differed significantly between Dragonfly and either microscopy or RDT. McNemar’s test is specifically suited for paired binary data, as it assesses asymmetry in discordant pairs (*i.e.*, instances where one method detects an infection while the other does not). In contrast, Cohen’s kappa measures overall agreement between methods but does not assess systematic differences in detection performance.

Reviewer #4 (Remarks to the Author):

We thank the reviewer’s time and energy in providing the valued feedback that we hope we have addressed sufficiently above.

Response to reviewers' comments:

Reviewer #1 (Remarks to the Author):

The paper is substantially improved with much more data, more impact, and more perspective. Nice work by the authors to make thoughtful revisions. I have a few additional comments and suggestions:

We are delighted to hear that Reviewer #1 finds the revised manuscript substantially improved, and we appreciate the positive feedback.

1. It may be useful to approximate the volume that is captured by the DBS-qPCR sample (3 discs of 3 mm diameter = ~9 uL blood) to add to the perspective on the impact of sample volume on LoD.

This information has now been added to the manuscript and underlined in the relevant sections, lines 191 (Results) and 342 (Methods).

2. It seems like you could write *P. falciparum* after the first instance of Plasmodium falciparum and save some space as the full name is written out a lot.

Thank you for the suggestion, the abbreviation *P. falciparum* has now been added to the manuscript.

3. Lines 70-72 should also include reverse transcription PCR methods (PMIDs: 40461818, 38185134, 31017084) and RT-LAMP method (PMIDs: 33508016, 37041454) for completeness.

This has now been added to the introductory section.

4. Supplemental Table 4: Use "x" signs for dimensions (not *) for consistency. Also use decimal points or commas consistently for weight depending on journal standards. Check the other spacing in this table as there's some typographical errors and sloppy inconsistencies

Table 4 has been edited to keep it consistent.

5. Over time, I would suggest that you develop something better than hand labelling the tube tops as a tracking system. 2D barcodes abound, and cell phone-based or other QR code scanner apps could be applied. Boring but important parts like this sample identification/labelling consideration are important when talking about scaling. It's not necessary from my perspective to expand on this in the paper herein, but I share this as guidance for ongoing implementation and roll-out of this strategy.

We appreciate the reviewer's suggestion regarding the development of a more sophisticated sample tracking system, and we agree that robust sample identification and labelling are critical for successful implementation and rollout. We have explored such labelling methodologies in previous studies (<https://www.nature.com/articles/s41467-025-57647-3>). However, our primary consideration for the current prototype iteration described in this manuscript, particularly for deployment in resource-limited settings, has been to prioritize simplicity, cost-effectiveness, and

operational feasibility. Implementing a system reliant on 2D barcodes, scanners, and potentially dedicated devices would introduce a more complex and potentially less accessible infrastructure requirement leading to a cost increase. We will continue to evaluate and improve the sample identification/labelling integration in future iterations as technology becomes more ubiquitous and affordable in these regions.

6. The following lines 377-379 are not quite right since there is no WB-qPCR for the 672 capillary samples in the Supplementary data ("Detailed results obtained using Dragonfly, Alethia®, DBS-qPCR, and WB-qPCR for each of the 50 positive controls from confirmed malaria patients, as well as the 672 capillary blood samples, are provided in the Supplementary Data."). I don't see any WB-qPCR in the Supplementary File so best to either add the data if available or remove/edit the comment. I'm surprised that the specificity of your assay didn't suffer against DBS-qPCR since your DF LoD is better than the DBS-qPCR.

We thank the Reviewer for their careful reading and valuable observations. The Reviewer is correct regarding lines 377–379. This was an error in the text, and we have now corrected it. The capillary blood samples were not tested using WB-qPCR, and we have updated the sentence accordingly.

Regarding the comment on specificity, we agree that it was initially unexpected that our assay's specificity did not show a greater decrease compared to DBS-qPCR, given the improved LoD of the Dragonfly assay. Our hypothesis is that this may be due to the relatively small number of samples with parasite concentrations between 0.6 (Dragonfly's LoD) and 2.9 (DBS-qPCR's LoD) parasites per microlitre. However, the 17 false positives observed (negative by microscopy, expert LM, and DBS-qPCR) resulting in a specificity of 96.8%, suggest that some reduction in specificity did occur. Unfortunately, further investigation is limited by the lack of paired whole blood samples collected in the field. This is consistent with the higher sensitivity of the assay. In previous evaluations of the Dragonfly platform using other targets such as SARS-CoV-2 and OPXV/MPXV, specificity relative to standard extracted qPCR has consistently been high, often reaching 99% or higher.

Reviewer #2 (Remarks to the Author):

This is the second revision of the manuscript entitled "High-sensitivity near point-of-care detection of asymptomatic and sub-microscopic Plasmodium infections in African endemic countries". The authors have deeply revised and much improved their original manuscript. They have strengthened their results by adding more tested samples (submicroscopic parasitaemia) and conducting new experiments with comparison of analytical performance of different methods (Figure 4). The latter were performed with the collaboration of researchers at the School of Hygiene & Tropical Medicine and the Imperial College of London with the obvious need to add two coauthors (Lindsay B Stewart and Pantelis Georgiou). I think that the suggested revisions have been formally addressed, and I would like to congratulate the authors on their significant efforts to do it.

We are very pleased to hear that Reviewer #2 finds the suggested revisions have been addressed satisfactorily, and we appreciate the positive feedback on our efforts to improve the manuscript.

1. I know very well that *P. falciparum* is the main Plasmodium species in sub-Saharan Africa, and you used only *P. falciparum* positive samples in the study. Accordingly, add *falciparum* in the title.

We have now added *falciparum* in the title.

2. Although the figures 1-3 are explicative I wonder if a short video clip can be added to the supplementary materials (if feasible).

While we do have an explanatory video that illustrates the core principles of the SmartLid extraction kit, it describes a protocol specifically designed for viral nucleic acid extraction from cell-free or near cell-free sample media (<https://www.youtube.com/watch?v=12q0etnTMGE>). Therefore, to avoid potential confusion for readers of this manuscript, we have opted not to include it.

©2023 ProtonDx Ltd. All rights reserved.

3. References: the name of microorganism should be in italic with the lower case of the second name.

This has now been addressed.

Reviewer #3 (Remarks to the Author):

The authors have satisfactorily addressed most of our previous comments. The authors collected more samples from asymptomatic individuals which allowed them to highlight the sensitivity difference between Dragonfly (95%) and expert microscopy (71%). qPCR results confirmed that the samples included a broad range of parasite concentrations, including many at sub-microscopic concentrations. This is important because detecting sub-microscopic levels of the parasite is one of the advantages of Dragonfly. The authors also tested a large number of negative controls to address and disprove a possible false positive issue. Altogether, these significant changes make the article more impactful.

We thank Reviewer #3 for the positive feedback.

A few final requests (numbers correspond to items in response to reviewers):

1. Add length of time and temps that lyophilized reagents are stable for in the discussion

This has been added in lines 293-296: “For example, studying the practical shelf-life of the test kits at a realistic range of room-temperatures throughout the continent will be critical. Current prototype Dragonfly test panels are conservatively labelled to expire after at least six months at 30°C, while the SmartLid Blood DNA/RNA Extraction Kits remains stable for at least 12 months.”

2. If Pan & Pf primers are in same tube, how do you distinguish which target is present? Is that not important? Please explain the rationale (in the manuscript) for having both targets in the same tube and the inability to distinguish whether the positive is from *P. falciparum* or broad Pan detection. Since the Pan primers in principle should detect *P. falciparum*, what is the benefit of including the Pf-specific primers in addition to the Pan primers? Is it intended to improve sensitivity for *P. falciparum* since this is the most prevalent species in the region where the study was conducted? And would a different strategy be appropriate (e.g. Pan only, or a different mixture of primers) in regions where a different species is more prevalent?

We thank the Reviewer for this comment and for highlighting an important point. We have now made our rationale clearer in lines 160–162 of the revised manuscript: “Primers specifically targeting *P. falciparum* were included in the reaction to improve the overall sensitivity of the test, given that *P. falciparum* is the most prevalent species in the region where the study was conducted.”

The main reason for combining both pan-*Plasmodium* and *P. falciparum*-specific primers in the same tube is to improve the sensitivity for detecting *P. falciparum*. This species is the most common in the region where the study was carried out. Although the pan primers are designed to detect *P. falciparum* along with other *Plasmodium* species, the addition of Pf-specific primers increases the chance of detecting low-density *P. falciparum* infections. These infections are often missed by microscopy or rapid diagnostic tests and are important in terms of ongoing transmission.

We agree with the Reviewer that in regions where *P. falciparum* is not the dominant species, a different primer strategy might be more appropriate. In areas where *P. vivax*, *P. ovale*, or *P. malariae* are more prevalent, or where species-level differentiation is important for treatment decisions, an assay designed to distinguish between species would be more suitable. Our current assay is tailored to the epidemiological profile of many sub-Saharan African countries, but future versions could be adapted for other regions based on local species distribution.

3. Please include text in the methods that even an “intermediate color shift (from the color reference control)” will be considered a positive test result.

This has now been added in lines 409-410: “Note, intermediate colour shifts (from pink to yellow) were considered as positive results, as this still indicated amplification of product.”

4. Please add discussion as to why microscopy isn't practical – not scalable, time-consuming, personnel constraints. There were some great points in the response to reviewers, but no text was added to the manuscript.

This has now been explained in lines 241-246, with a focus on the main arguments from the previous response to reviewers as suggested: "This rigorous approach likely contributed to the higher sensitivity by LM observed in our study compared to routine clinical practice, where blood slides may be read by less experienced technicians. Furthermore, despite the comparable performance of LM for parasite densities above 16 parasites/ μ L demonstrated through this study, multiple practical limitations remain that make LM unsuitable for large-scale asymptomatic screening. For example, the preparation and reading of a slide, particularly at low parasite densities, can take an experienced microscopist at least 25-30 minutes per sample. In contrast, the presented method can deliver results for up to 12 samples simultaneously in an average of 45 minutes per minimally trained operator."

Reviewer #4 (Remarks to the Author):

We would like to thank Reviewer #4 for the time and effort invested in reviewing our manuscript.